# The Logical Expressiveness of Temporal GNNs via Two-Dimensional Product Logics

**Marco Sälzer**
RPTU Kaiserslautern-Landau
Kaiserslautern, Germany
`marco.saelzer@rptu.de`

**Przemysław Andrzej Wałęga**
Queen Mary University of London, UK
University of Łódź, Poland
`p.walega@qmul.ac.uk`

**Martin Lange**
Theoretical Computer Science / Formal Methods
University of Kassel, Germany
`martin.lange@uni-kassel.de`

## Abstract

In recent years, the expressive power of various neural architectures—including graph neural networks (GNNs), transformers, and recurrent neural networks—has been characterised using tools from logic and formal language theory. As the capabilities of basic architectures are becoming well understood, increasing attention is turning to models that combine multiple architectural paradigms. Among them particularly important, and challenging to analyse, are temporal extensions of GNNs, which integrate both spatial (graph-structure) and temporal (evolution over time) dimensions. In this paper, we initiate the study of logical characterisation of temporal GNNs by connecting them to two-dimensional product logics. We show that the expressive power of temporal GNNs depends on how graph and temporal components are combined. In particular, temporal GNNs that apply static GNNs recursively over time can capture all properties definable in the product logic of (past) propositional temporal logic PTL and the modal logic K. In contrast, architectures such as graph-and-time TGNNs and global TGNNs can only express restricted fragments of this logic, where the interaction between temporal and spatial operators is syntactically constrained. These provide us with the first results on the logical expressiveness of temporal GNNs.

## 1 Introduction

Recent years have seen significant progress in understanding the expressive power of neural architectures using tools from logic, formal language theory, and graph theory. Some of the most prominent results concern *Graph Neural Networks* (GNNs), whose distinguishing power has been famously characterised with the Weisfeiler-Leman isomorphism test [31, 46] and whose logical expressiveness has been captured with modal and first-order logics [3, 17, 32, 5, 1]. These insights have revealed both the limitations and strengths of GNNs, inspiring the development of more expressive variants such as higher-order GNNs [31]. They have also opened the way for extracting logical rules from GNNs, advancing explainability in graph-based learning [42, 43]. Consequently, the logical expressiveness of GNNs has become a rapidly evolving research area.

As the capabilities of standard GNN architectures become relatively well understood, research interest is shifting to more complex architectures, which combine multiple dimensions of structure. One particularly prominent and challenging case is that of *Temporal Graph Neural Networks* (TGNNs) [26, 40, 14], which can be seen as an extension of GNNs enabling to process *temporal graphs*,

39th Conference on Neural Information Processing Systems (NeurIPS 2025).

that is, graphs whose topology evolves over time. As a result, TGNN computations combine the spatial (graph structure) with temporal (changes in time) dimensions. Considerable advancements have been made in the design of various TGNN architectures [26] and their deployment across diverse applications such as traffic forecasting, financial applications, and epidemiological contexts [49, 33, 21]. However understanding expressive capabilities of TGNNs remains limited, with the first steps only comparing some TGNN architectures [14, 8] or establishing relations with temporal versions of the Weisfeiler-Leman test [41, 44]. To the best of our knowledge no connection of TGNNs with logics has been established prior to this work.

**Our contribution.** In this work, we initiate the analysis of the logical expressiveness of TGNNs. Our goal is to characterise the temporal graph properties that TGNNs can express using logical languages. To achieve this, we propose a novel approach—analysing the logical expressiveness of TGNNs using two-dimensional product logics. This is based on the key insight that the computations of TGNNs, which process both the structural properties of static graphs and their evolution over time, naturally correspond to combinations of modal and temporal properties.

Both modal and temporal logics have been extensively studied in the logic community for decades, and more recent work has connected them to the expressiveness of neural architectures [3, 32, 4, 47]. Likewise, many-dimensional logics, especially products of modal and temporal logics, have a rich theoretical foundation and well-established tools for analysing definability and complexity [24, 6, 28]. We are the first to apply product logics as a framework for analysing the expressive power of TGNNs.

At a technical level, we show that certain TGNN architectures are (in-)capable of expressing certain combinations of modal-temporal properties. Our results are as follows:

- The class $\mathcal{T}_{\mathsf{rec}}[\hat{\mathcal{M}}]$ of TGNNs recursively applying standard static GNNs (from class $\hat{\mathcal{M}}$) over time, can express all properties definable in the product logic $\mathsf{PTL}_{\mathsf{P,Y}} \times \mathsf{K}$, which combines the seminal (past) propositional temporal logic $\mathsf{PTL}_{\mathsf{P,Y}}$ and the modal logic $\mathsf{K}$ (Theorem 6).

- Analogous results hold if we replace $\hat{\mathcal{M}}$ and $\mathsf{K}$ with matching expressiveness GNNs and logics. In particular, we show that such results hold for two specific pairs of static GNNs and logics, recently studied in literature [32, 5] (Theorem 7).

- In contrast to recursive TGNNs, the class of time-and-graph TGNNs [14] does not allow us to express all properties expressible in $\mathsf{PTL}_{\mathsf{P,Y}} \times \mathsf{K}$ (Theorem 8). We show, however, that time-and-graph TGNNs can express all properties definable in a fragment of $\mathsf{PTL}_{\mathsf{P,Y}} \times \mathsf{K}$, in which the allowed interplay between temporal and modal operators is syntactically restricted (Theorem 9).

- The class of global TGNNs [44] also does not allow us to express all properties expressible in $\mathsf{PTL}_{\mathsf{P,Y}} \times \mathsf{K}$ (Theorem 10). As in the previous case, we determine a fragment of $\mathsf{PTL}_{\mathsf{P,Y}} \times \mathsf{K}$, such that global TGNNs can express all properties definable in this fragment (Theorem 11). This fragment, however, allows for different interactions between temporal and modal operators, than the fragment from Theorem 8.

- We show how our results allow to determine relative expressive power of TGNN classes considered in the paper (Corollary 12 and Theorem 13).

Beyond their theoretical significance, these results open the door to novel avenues in explainable and trustworthy AI. Just as logical characterisations of static GNNs have enabled the extraction of symbolic rules from neural models [42, 43], our analysis lays the foundation for extracting temporal rules and dynamic specifications from TGNNs.

## 2 Related work

**Temporal GNNs.** There exists a plethora of TGNN models [26], differing on representation and processing temporal graphs. There exist various classifications of TGNNs [26], but one of the main distinction is between snapshot-based models, where a temporal graph is given as a sequence of its timestamped snapshots [18, 38, 30, 48, 11], and event-based models, where a temporal graph is given as a sequence of events modifying the graph structure [45, 35, 27]. In this paper we focus on the snapshot-based TGNNs, but it is worth to observe that in many settings these two types of temporal graph representation can be translated into each other. Initial work on the expressive power of TGNNs has been conducted exploiting their relation to variants of the Weisfeiler-Leman isomorphism test

[41, 44] and by directly comparing expressiveness of particular TGNN architectures [14, 8]. However, to the best of our knowledge, no work has yet studied the expressive power of TGNNs from the perspective of logics.

**Logical expressiveness of static GNNs.** The seminal logical characterisation of (message-passing) GNNs established that the properties expressible both by GNNs and first-order logic coincide with those definable in graded modal logic [3]. Subsequent work moved beyond the assumption of first-order expressibility, aiming to identify logics that capture the full expressive power of GNNs. This line of research led to the introduction of logics such as the modal logic $\mathsf{K}^{\#}$ [32] and logics extended with Presburger quantifiers [5]. More recent results have provided logical characterisations of further GNN architectures using logics with quantised parameters [1, 37] or standard modal logics [12]. There are also results on the logical expressiveness of GNNs extended with global readouts [3, 20].

**Product logics** Multi-dimensional product logics [24, 29] offer a general framework for combining multiple modal logics, each capturing distinct aspect of reasoning such as space, time, or knowledge. In these systems, states are represented as tuples drawn from the component logics, and accessibility relations are defined componentwise, enabling interaction between different modal dimensions. There is a long-standing tradition of studying the complexity and axiomatisation of multi-dimensional logics—a line of research that often proves to be highly challenging [24, 29]. Two-dimensional product logics, in particular, have been extensively investigated due to their applicability in spatio-temporal [6], temporal-epistemic [19], temporal-standpoint [15, 13], and temporal description logic [2] settings. Of particular relevance to our work is the two-dimensional product logic $\mathsf{PTL} \times \mathsf{K}$, which combines propositional temporal logic (PTL) [34] with the modal logic $\mathsf{K}$ [23]. This logic, along with its various extensions and variants, has been the subject of extensive study in the literature [24].

## 3   Notation

We will briefly describe basic notions, notation, and conventions used in the paper.

We use bold symbols, such as $\boldsymbol{x}$ and $\boldsymbol{y}$, to represent vectors; we assume that they are given as column vectors. In particular, we use $\boldsymbol{0}$ for vectors containing only 0s; their dimension will usually be clear from context. We let $\boldsymbol{x}_i$ be the $i$th element of $\boldsymbol{x}$, and $\boldsymbol{x} \,\|\, \boldsymbol{y}$ the concatenation of $\boldsymbol{x}$ and $\boldsymbol{y}$, that is, a vector obtained by stacking $\boldsymbol{x}$ onto $\boldsymbol{y}$. We use $\{\!\{\cdots\}\!\}$ to denote multisets, that is, sets with possibly multiple occurrences of elements.

A (static, undirected, node-labelled) *graph*, $G = (V, E, c)$, is a triple consisting of a finite set $V$ of nodes, a set $E$ of undirected edges over $V$, and a labelling $c : V \to \mathbb{R}^k$ of nodes with vectors in $\mathbb{R}^k$. If $c : V \to \{0, 1\}^k$, we call vector entries *colours* and say that the graph is *coloured* (with $k$ colours). A *pointed graph* is a pair $(G, v)$ consisting of a graph and one of its nodes.

A *temporal graph*, $TG = (G_1, t_1), \ldots, (G_n, t_n)$, is a finite sequence of pairs $(G_i, t_i)$, where $G_i = (V_i, E_i, c_i)$ is a static graph such that $V_1 = \cdots = V_n$ and $t_i \in \mathbb{R}$ is a number, also called a *timestamp*, such that $t_1 < \cdots < t_n$. We call $n$ the *length* of $TG$. We usually refer to the set of nodes of $TG$ as $V$. We call a temporal graph *discrete*, if $t_i = i$ for each timestamp $t_i$. A *timestamped node* is a pair $(v, t_i)$ of a node and a timestamp; a *pointed temporal graph* is a pair $(TG, (v, t_i))$ of a temporal graph and a timestamped node. Often, we will write $(TG, v)$ instead of $(TG, (v, t_n))$.

## 4   Message-passing GNNs and TGNNs

In this section, we briefly present the standard message-passing GNNs [16], referred to as message passing neural networks (MPNNs) here, but also called aggregation-combine GNNs (AC-GNNs) [3]. Afterwards, we present three ways of extending them to the temporal setting, which gives rise to the three classes of temporal GNNs we will study in this paper.

**Definition 1.** *A* message-passing GNN *(MPNNs),* $M = (l^1, \ldots, l^k)$, *is a finite sequence of* message-passing layers *of the form* $l^i = (\mathsf{comb}^i, \mathsf{agg}^i)$, *where* $\mathsf{comb}^i$ *are* combination *functions mapping pairs of vectors to single vectors, and* $\mathsf{agg}^i$ *are* aggregation *functions mapping multisets of vectors to single vectors. An application of $M$ to a graph $G = (V, E, c)$ yields embeddings $\mathbf{h}_v^{(i)}$ computed for*

*all $v \in V$ and $i + 1 \leq k$ as follows:*

$$\mathbf{h}_v^{(0)} = c(v), \qquad \mathbf{h}_v^{(i+1)} = \mathsf{comb}^{i+1}(\mathbf{h}_v^{(i)}, \mathsf{agg}^{i+1}(\{\!\{\mathbf{h}_u^{(i)} \mid \{v, u\} \in E\}\!\})).$$

*For a pointed graph $(G, v)$, we let $M(G, v) = \mathbf{h}_v^{(k)}$ be the final embedding computed for $v$.*

Depending on the type of combination and aggregation functions we obtain various classes of MPNNs, usually written as $\mathcal{M}$. In particular, we use $\hat{\mathcal{M}}$ for the class of MPNNs whose combination functions are realised by feedforward neural networks with truncated-ReLU $\max(0, \min(1, x))$ as activation function and with aggregation given by the entrywise sum $\mathsf{agg}(S) = \sum_{\boldsymbol{x} \in S} \boldsymbol{x}$. We denote the class of these FNN by $\mathsf{FNN}[\mathsf{trReLU}]$ (see Appendix A.1 for a formal definition). Another class of MPNNs we consider is $\hat{\mathcal{M}}_{\mathsf{msg}}$ which is similar to $\hat{\mathcal{M}}$, but aggregations are now given by $\mathsf{agg}(S) = \sum_{\boldsymbol{x} \in S} \mathsf{msg}(\boldsymbol{x})$, where $\mathsf{msg}$ is a feedforward neural network from $\mathsf{FNN}[\mathsf{trReLU}]$.

Next, we discuss three classes of TGNNs obtained by extending MPNNs in order to process temporal graphs $TG = (G_1, t_1), \ldots, (G_n, t_n)$. The first type, recursive TGNNs, starts processing $TG$ by applying an MPNN to the first static graph $G_1$. It then extends $G_2$ by concatenating node labels with the embeddings computed by the MPNN. Next, it applies the same MPNN to the obtained static graph. This process is applied recursively in $n$ rounds, until all graphs $G_1, \ldots, G_n$ are processed. The formal definition of such models is given below.

**Definition 2.** Recursive TGNNs $\mathcal{T}_{\mathsf{rec}}[\mathcal{M}]$, *for a class $\mathcal{M}$ of MPNNs, are pairs $T = (M, \mathsf{out})$, where $M \in \mathcal{M}$ and $\mathsf{out}$ is an output function mapping vectors to binary values in $\{0, 1\}$. An application of $T = (M, \mathsf{out})$ to a temporal graph $TG = (G_1, t_1), \ldots, (G_n, t_n)$ with $G_j = (V_j, E_j, c_j)$, yields the following embeddings for all $v \in V$ and $j + 1 \leq n$:*

$$\mathbf{h}_v^{(0)}(t_1) = c_1(v) \,\|\, \mathbf{0}, \qquad \mathbf{h}_v^{(0)}(t_{j+1}) = c_{j+1}(v) \,\|\, \mathbf{h}_v^{(k)}(t_j),$$
$$\mathbf{h}_v^{(k)}(t_j) = M((V_j, E_j, [u \mapsto \mathbf{h}_u^{(0)}(t_j)]), v).$$

*where $k$ is the number of layers in $M$ and $[u \mapsto \mathbf{h}_u^{(0)}(t_j)]$ is a labelling function mapping each $u \in V_j$ to the vector $\mathbf{h}_u^{(0)}(t_j)$,*

We remark that we introduce the $\mathcal{T}_{\mathsf{rec}}$ architecture to represent approaches that directly leverage existing MPNN architectures for temporal graphs [48] without employing a specialised architecture.

The second type, time-and-graph TGNNs, are studied in several papers [14, 8] and akin to several models [25, 7, 39]. Such TGNNs perform computations by exploiting not only MPNNs, but also *cell functions* (e.g. a gated recurrent unit [10]) mapping vectors to vectors. In particular, time-and-graph TGNNs use two MPNNs ($M_1$ and $M_2$) and one cell function ($Cell$). A temporal graph $TG = (G_1, t_1), \ldots, (G_n, t_n)$ is, again, processed from left to right in $n$ steps. In each step $j + 1$, the TGNN applies $M_1$ to $G_{j+1}$ and $M_2$ to $G_{j+1}$ with node labels replaced by embeddings computed in step $j$. This results in computing two vectors for each node, which are combined into a single vector using $Cell$. After $n$ steps of such processing, the TGNN terminates its computations.

**Definition 3.** Time-and-graph *TGNNs* $\mathcal{T}_{\mathsf{TandG}}[\mathcal{M}, \mathcal{C}]$, *for a class $\mathcal{M}$ of MPNNs and a class $\mathcal{C}$ of cell functions, are tuples $T = (M_1, M_2, Cell, \mathsf{out})$, where $M_1, M_2 \in \mathcal{M}$ have the same number of layers, $Cell \in \mathcal{C}$, and $\mathsf{out}$ is an output function (as in Definition 2). An application of $T = (M_1, M_2, Cell, \mathsf{out})$ to a temporal graph $TG = (G_1, t_1), \ldots, (G_n, t_n)$ with $G_j = (V_j, E_j, c_j)$, yields the following embeddings for all $v \in V$ and $j + 1 \leq n$:*

$$\mathbf{h}_v(t_1) = Cell(M_1(G_1, v), M_2((V_1, E_1, [u \mapsto \mathbf{0}]), v)),$$
$$\mathbf{h}_v(t_{j+1}) = Cell(M_1(G_{j+1}, v), M_2((V_{j+1}, E_{j+1}, [u \mapsto \mathbf{h}_u(t_j)]), v)),$$

*where $[u \mapsto \mathbf{h}_u(t_j)]$ is as described in Definition 2.*

The third class, global TGNNs [44], is also realised by several TGNN models [35, 27]. Instead of processing a temporal graph in each time point, global TGNNs exploit a temporal message-passing mechanism. It allows messages to be passed among nodes from different time points. To capture the time difference between nodes between which messages are passed, global TGNNs use *time functions* $\phi : \mathbb{R} \to \mathbb{R}^m$.

**Definition 4.** Global *TGNNs* $\mathcal{T}_{\mathsf{glob}}[\mathcal{M}, \mathcal{Q}, \circ]$, *for a class $\mathcal{M}$ of MPNNs, a class $\mathcal{Q}$ of time functions, and an operation $\circ$ combining two vectors of the same dimensionality, are tuples $T = (M, \phi, \mathsf{out})$*

*where $M \in \mathcal{M}$ $\phi \in \mathcal{Q}$, and* out *is an output function (as in Definition 2). An application of* $T = (M, \phi, \text{out})$ *to a temporal graph* $TG = (G_1, t_1), \ldots, (G_n, t_n)$ *with* $G_j = (V_j, E_j, c_j)$, *yields embeddings* $\mathbf{h}_v^{(i+1)}(t_j)$ *for all* $v \in V$, $i + 1 \leq k$ *(for $k$ the number of layers in $M$), and $j \leq n$:*

$$\mathbf{h}_v^{(0)}(t_j) = c_j(v),$$
$$\mathbf{h}_v^{(i+1)}(t_j) = \mathsf{comb}^{i+1}(\mathbf{h}_v^{(i)}(t_j), \mathsf{agg}^{i+1}\{\!\{\mathbf{h}_u^{(i)}(t_h) \circ \phi(t_j - t_h) \mid \{v, u\} \in E_h, h \leq j\}\!\}).$$

In all TGNN models considered above, an output function out is used to determine the final classification. Formally, for a TGNN $T$ and a pointed temporal graph $(TG, v)$, we let the output of $T$ be $T(TG, v) = \mathsf{out}(\mathbf{h}_v^{(k)}(t_n))$. We will assume that out functions are realised by a single-layer FNN from the class FNN[trReLU].

## 5  Logical Expressiveness via Product Logics

We will exploit two-dimensional-product logics as a tool for analysing the expressive power of TGNNs. Specifically, we consider two-dimensional logics which are products of modal and temporal logics, allowing combinations of structural and temporal properties to be expressed. Importantly, this approach provides an opportunity to control the interaction between logical operators corresponding to the two dimensions. By modifying allowed interactions in product logics we will obtain logics that are suitable to analyse the expressive power of various classes of TGNNs.

Before introducing product logics, we specify the notion of the expressive power that we will consider in this paper. It corresponds to the "logical expressiveness" also called "uniform expressiveness", which is broadly studied in the literature [3, 20, 32, 5, 12] and is defined as follows.

**Definition 5.** *Let $\mathcal{L}$ be a logical language, whose formulas $\varphi$ are evaluated at timestamped nodes $(v, t_n)$ of temporal graphs $TG = (G_1, t_1), \ldots, (G_n, t_n)$. We write $TG, (v, t_n) \models \varphi$ if $\varphi$ holds at $(v, t_n)$ in $TG$. A TGNN class $\mathcal{T}$ is at least as expressive as $\mathcal{L}$, written $\mathcal{L} \leq \mathcal{T}$, if for every formula $\varphi$ of $\mathcal{L}$, there exists a model $T \in \mathcal{T}$ such that, for all (coloured) pointed temporal graphs $(TG, (v, t_n))$:*

$$TG, (v, t_n) \models \varphi \text{ if and only if } T(TG, v) = 1.$$

Informally, $\mathcal{L} \leq \mathcal{T}$ means that TGNNs from the class $\mathcal{T}$ are powerful enough to express all properties that can be written as (arbitrarily long and complex) formulas of the logics $\mathcal{L}$.

In order to study the expressive power of TGNNs, we will establish their relation to two-dimensional product logics [24, 6, 28]. The first dimension will correspond to time and will allow us to capture evolution of a graph in time. The second dimension will correspond to the spatial structure of graphs and will allow us to express properties of the static graphs at fixed time points. The product logics we consider here are combinations of the most prominent temporal and modal logics, namely *propositional temporal logic* PTL [34] and the *basic modal logic* K [23].

**Temporal logic $\mathsf{PTL_{P,Y}}$.**  We consider the past-time version $\mathsf{PTL_{P,Y}}$ of PTL which contains only P ("*sometime in the past*") and Y ("*yesterday*") as native temporal operators. $\mathsf{PTL_{P,Y}}$ formulas are built using propositions $c_1, \ldots, c_k$ which stand for colours that nodes of temporal graphs can take, together with an unrestricted use of Boolean connectives ¬ for "not", ∧ for "and", → for "if ... then ...", and ↔ for " if and only if". Formulas of the logic $\mathsf{PTL_{P,Y}}$ can state, for example, that a node has currently colour $c_1$ and sometime in the past it had colour $c_2$ in two consecutive timepoints. This is written as $c_1 \wedge \mathsf{P}(c_2 \wedge \mathsf{Y}c_2)$.

**Modal logic K.**  For the static part we use the basic modal logic K. Its formulas are similar to those of $\mathsf{PTL_{P,Y}}$, but instead of the temporal operators P and Y it features a single modal operator $\Diamond$. The intuitive reading of $\Diamond\varphi$ is "*$\varphi$ holds at one of neighbours (via graph edges) of the current node*". Logic K can express, for example, that a node has an outgoing path of three hops leading to a node of colour $c_1$ or a two-hop path to a node whose colour is $c_2$ and not $c_3$: $\Diamond\Diamond\Diamond c_1 \vee \Diamond\Diamond(c_2 \wedge \neg c_3)$.

**Product logic $\mathsf{PTL_{P,Y}} \times \mathsf{K}$.**  The product of logics $\mathsf{PTL_{P,Y}}$ and K allows us to write formulas with any combination of operators from $\mathsf{PTL_{P,Y}}$ and K. For example, $\varphi = c_1 \wedge (\mathsf{P}c_2) \wedge \Diamond((\neg c_1 \wedge c_2) \wedge \mathsf{Y}(c_1 \wedge \neg c_2))$ is satisfied at a vertex $v$ in time point $t_n$ if $v$ is coloured $c_1$ at $t_n$ and at some past

timestamp $t_i < t_n$, $v$ was coloured with $c_2$. Moreover, there is a neighbour of $v$ that is currently coloured with $c_2$ but not $c_1$ at $t_n$, whereas in the preceding timestamp $t_{n-1}$ it was coloured with $c_1$ but not with $c_2$. An example of a pointed temporal graph $(TG, v)$ satisfying $\varphi$ in $t_n = t_4$ is given in Figure 1. From the technical perspective, the semantics of $\mathsf{PTL_{P,Y} \times K}$ is defined over Cartesian products of models of $\mathsf{PTL_{P,Y}}$ and $\mathsf{K}$, which justifies the name "product logic". A formal definition of syntax and semantics of this product logic is given in Appendix A.2.

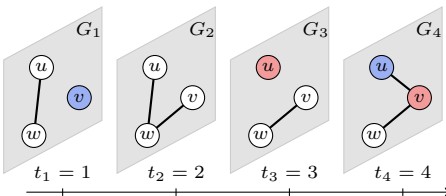

Figure 1: A temporal graph of length 4; colour $c_1$ is denoted by a red filling (node $u$ in $G_3$ and node $v$ in $G_4$) and colour $c_2$ is denoted by a blue filling (node $v$ in $G_1$ and node $u$ in $G_4$)

We also consider variants of $\mathsf{PTL_{P,Y} \times K}$ obtained by replacing the modal logic $\mathsf{K}$ with other modal logics known from the literature, or by syntactically restricting how temporal and modal operators can co-occur in formulas. Such variants differentiate the expressive power of various TGNN models.

## 6 Logical expressiveness of recursive TGNNs

We provide expressive power results for recursive TGNNs by relating them with product logics. We show that TGNNs from the class $\mathcal{T}_{\mathsf{rec}}[\hat{\mathcal{M}}]$ can express all properties definable in $\mathsf{PTL_{P,Y} \times K}$.

**Theorem 6.** $\mathsf{PTL_{P,Y} \times K} \le \mathcal{T}_{rec}[\hat{\mathcal{M}}]$.

*Proof sketch.* For each formula $\varphi \in \mathsf{PTL_{P,Y} \times K}$, we enumerate its subformulas $\varphi_1, \ldots, \varphi_m$, $\varphi_{m+1}, \ldots, \varphi_n = \varphi$ in such a way that the $m$ atomic ones come first, and if $\varphi_i$ is a subformula of $\varphi_j$, written as $\varphi_i \in sub(\varphi_j)$, then $i \le j$. We construct a TGNN $T_\varphi = (M, \mathsf{out}) \in \mathcal{T}_{\mathsf{rec}}[\hat{\mathcal{M}}]$ where $M$ consists of $n - m + 1$ layers: one layer for each non-atomic subformula and a final shift layer. At each time $t$ and for each node $u$ of the current temporal graph, $T_\varphi$ computes the hidden state $\mathbf{h}_u^{(n-m)}(t)$, which encodes (I) the current truth values of all subformulas $\varphi_i$ in the first $n$ dimensions, (II) the truth values of all subformulas $\varphi_i$ at $t - 1$ in dimensions $n + 1$ to $2n$, and (III) the disjunction of truth values of subformulas $\varphi_i$ over all earlier time points in dimensions $2n + 1$ to $3n$. The final shift layer (layer $n - m + 1$) updates (III) using (II) without altering (I). Correctness follows by nested induction over time and subformula structure, showing that for each $i \le n$, dimension $i$ correctly tracks satisfaction of $\varphi_i$, dimension $n + i$ tracks $\mathsf{Y}\varphi_i$, and dimension $2n + i$ tracks $\mathsf{P}\varphi_i$. The output $out$ simply reads the value for $\varphi$ at the final time point. A full proof is given in Appendix B.1. $\quad\square$

We provide two further analogous results obtained by replacing $\mathsf{K}$ and $\hat{\mathcal{M}}$ with pairs of logics and MPNNs of matching expressiveness known from the literature. This suggests a general connection between product logics and recursive TGNNs. First, $\mathsf{K}$ can be replaced with the logic $K^{\#}$, incorporating linear arithmetics, and $\hat{\mathcal{M}}$ with the class of MPNNs $\mathcal{M}_{K\#}$ of matching expressive power [32]. Second, an analogous result is obtained when using the logic $\mathcal{L}\text{-MP}^2$, incorporating Presburger Arithmetic, and the class $\mathcal{OL}\mathsf{trReLU\text{-}GNN}$ of MPNNs, for which matching expressiveness results have been shown [5]. For the formal details of these logics, see Appendix A.3, and we remark that it is evident that both strictly subsume $\mathsf{PTL_{P,Y} \times K}$.

**Theorem 7.** $\mathsf{PTL_{P,Y} \times} K^{\#} \le \mathcal{T}_{rec}[\mathcal{M}_{K\#}]$ *and* $\mathsf{PTL_{P,Y} \times}(\mathcal{L}\text{-MP}^2) \le \mathcal{T}_{rec}[\mathcal{OL}\mathsf{trReLU\text{-}GNN}]$.

*Proof sketch.* We apply similar inductive argument as used in Theorem 6. For each formula $\varphi$, we enumerate its subformulas $\varphi_1, \ldots, \varphi_{m_1}, \varphi_{m_1+1}, \ldots, \varphi_{m_k}, \varphi_{m_k+1}, \ldots, \varphi_n = \varphi$, so that all subformulas of the form $\mathsf{Y}\psi$ or $\mathsf{P}\psi$ occupy positions $m_i$ for $i = 1, \ldots, k$. Each group of purely non-temporal subformulas (situated between $m_i$ and $m_{i+1}$, assuming that formulas $\varphi_i$ with $i \le m_i$ are already addressed) is captured inductively by stacks of MPNN layers whose existence is guaranteed by results

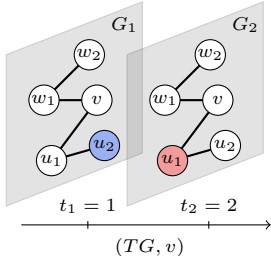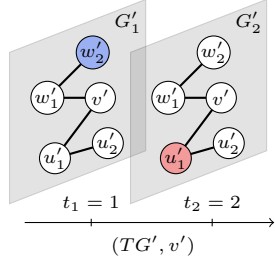

Figure 2: Counterexample, used in Theorem 8; colour $c_1$ is denoted by a red filling (node $u_1$ in $G_2$ and node $u'_1$ in $G'_2$) and colour $c_2$ is denoted by a blue filling (node $u_2$ in $G_1$ and node $w'_2$ in $G'_1$)

from literature [5, 32]. This necessitates that $\mathcal{OL}\text{trReLU-GNN}$ and $\mathcal{M}_{K\#}$ are closed under arbitrary but well-defined combinations of MPNN layers. Temporal subformulas $\mathsf{Y}\psi$ and $\mathsf{P}\psi$ are handled by simple dimension shifts, reading values stored at previous timepoints or across past histories, similar to Theorem 6. Here, we use the same form of hidden states, divided into three blocks tracking the current, previous, and past satisfaction of subformulas. Correctness follows by nested induction over time and subformula structure: for non-temporal subformulas, correctness can be shown exploiting results from literature [5, 32], while for temporal subformulas, it follows from the way in which the temporal operators $\mathsf{Y}$ and $\mathsf{P}$ are addressed. The full proof is in Appendix B.2. □

## 7 Logical expressiveness of time-and-graph and global TGNNs

Next, we focus on two variants of TGNNs that have been considered in the literature thus far, aiming to unveil certain differences in their logical expressiveness.

First, we examine time-and-graph TGNNs [14], denoted by $\mathcal{T}_{\mathsf{TandG}}[\mathcal{M}, \mathcal{C}]$ and parametrised by a class of MPNNs $\mathcal{M}$ and a class of cell functions $C$. These TGNNs differ from the previously discussed classes $\mathcal{T}_{\mathsf{rec}}[\mathcal{M}]$, not in the specific form of MPNNs used, but in how temporal information is processed. In particular, $\mathcal{T}_{\mathsf{TandG}}$ appears less powerful at processing the combined information of a current snapshot $(G_i, t_i)$ and the embedded information of the past snapshots $(G_1, t_1), \ldots, (G_{i-1}, t_{i-1})$.

**Theorem 8.** $\mathsf{PTL}_{\mathsf{P},\mathsf{Y}}\times\mathsf{K} \not\leq \mathcal{T}_{\mathsf{TandG}}[\mathcal{M}, \mathcal{C}]$, for all $\mathcal{M}$ and $\mathcal{C}$.

*Proof sketch.* Consider the $\mathsf{PTL}_{\mathsf{P},\mathsf{Y}}\times\mathsf{K}$ formula $\varphi = \Diamond(c_1 \wedge \mathsf{Y}\Diamond c_2)$, satisfied by all pointed temporal graphs $(TG, (v, t_n))$ where $n \geq 2$ and node $v$ has a neighbour that is of colour $c_1$ at timestamp $t_n$ and which has a neighbour of colour $c_2$ at timestamp $t_{n-1}$. For example, see Figure 2, where $(TG, v)$ satisfies $\varphi$ and $(TG', v')$ does not. We can show that there exists no TGNN $T = (M_1, M_2, Cell) \in \mathcal{T}_{\mathsf{TandG}}[\mathcal{M}, \mathcal{C}]$ for any class of MPNNs $\mathcal{M}$ and cell functions $\mathcal{C}$ that captures $\varphi$. Intuitively, $M_1$ handles the present label information and $M_2$ the past information, but neither handles both. While $Cell$ can use outputs of $M_1$ and $M_2$, it has no access to the topology of the static graph. However, to check whether $\varphi$ is satisfied, this is necessary. This results in the fact that $T$ either accepts both $(TG, v)$ and $(TG', v')$ of Figure 2, or none. A formal proof of this is provided in Appendix B.3. □

The natural next question is to identify a fragment of $\mathsf{PTL}_{\mathsf{P},\mathsf{Y}}\times\mathsf{K}$ whose formulas can be expressed by time-and-graph TGNNs. Let $\mathcal{F} \subset \mathsf{FNN}[\mathsf{trReLU}]$ be the class of single layer FNNs with truncated-ReLU activations. We let $\mathcal{L}_1$ be the fragment of $\mathsf{PTL}_{\mathsf{P},\mathsf{Y}}\times\mathsf{K}$ containing only formulas $\varphi$ such that for all $\Diamond\psi \in sub(\varphi)$ there is either no $Q\chi \in sub(\psi)$ with $Q \in \{\mathsf{Y}, \mathsf{P}\}$ or for all $c \in sub(\psi)$ there is $Q\chi \in sub(\psi)$ with $Q \in \{\mathsf{Y}, \mathsf{P}\}$ such that $c \in sub(\chi)$. Hence, $\mathcal{L}_1$ restricts the allowed interaction of operators in $\mathsf{PTL}_{\mathsf{P},\mathsf{Y}}\times\mathsf{K}$. For example $\Diamond\mathsf{P}c_1$ is a formula of $\mathcal{L}_1$, but $\Diamond(\mathsf{P}c_1 \wedge c_2)$ is not.

**Theorem 9.** $\mathcal{L}_1 \leq \mathcal{T}_{\mathsf{TandG}}[\hat{\mathcal{M}}, \mathcal{F}]$.

*Proof sketch.* We apply the inductive approach of Theorems 6 and 7 to formulas $\varphi \in \mathcal{L}_1$. We construct time-and-graph TGNNs $T_\varphi = (M_1, M_2, Cell, \text{out}) \in \mathcal{T}_{\mathsf{TandG}}[\hat{\mathcal{M}}, \mathcal{F}]$ where the three components partition the evaluations according to the syntactic form of the subformulas $\varphi_i$ of $\varphi$. The MPNN $M_1$ captures all subformulas $\varphi_i$ that do not include the temporal operators $\mathsf{Y}$ or $\mathsf{P}$, meaning it processes static properties of the current snapshot. Similarly, the MPNN $M_2$ handles all subformulas

where every atomic subformula is nested under a temporal operator, indicating that the formula evaluation relies solely on the prior hidden states. The cell function $Cell$ manages the remaining subformulas, which must be Boolean subformulas ($\neg\psi_1$ or $\psi_1 \wedge \psi_2$) due to the definition of $\mathcal{L}_1$. Correctness is established through a nested induction over time and subformula structure, as done previously. The structure of $\mathcal{L}_1$ ensures that all subformulas are appropriately covered by the division among $M_1$, $M_2$, and $Cell$. A full proof is provided in Appendix B.4. $\qquad\square$

We remark that some time-and-graph TGNNs considered in the literature [14] utilise a gated recurrent unit (GRU) [9, 10] as their cell function. The class of TGNNs considered in Theorem 9 is a subset of this broader class. There are also similar, *time-then-graph* TGNNs, considered in the literature [14]. They first process the temporal information of a temporal graph using a recurrent neural network (RNN), followed by the application of an MPNN to capture the topology. It is shown that for each time-and-graph TGNN, there exists an equivalent time-then-graph TGNN [14, Theorems 3.5 and 3.6]. Thus, our results from Theorem 9 transfer to time-then-graph TGNNs.

Next we consider global TGNNs, $\mathcal{T}_{\text{glob}}[\mathcal{M}, \mathcal{Q}, \circ]$, recently studied in the literature [44]. They are parametrised by a class of MPNNs $\mathcal{M}$, time functions from $\mathcal{Q}$, and an operation $\circ$ combining label and temporal information in the aggregation. Similarly to time-and-graph TGNNs, $\mathcal{T}_{\text{glob}}$ cannot capture all properties definable in $\mathsf{PTL}_{\mathsf{P,Y}} \times \mathsf{K}$.

**Theorem 10.** $\mathsf{PTL}_{\mathsf{P,Y}} \times \mathsf{K} \not\leq \mathcal{T}_{\text{glob}}[\mathcal{M}, \mathcal{Q}, \circ]$, *for all $\mathcal{M}$, $\mathcal{Q}$, and $\circ$.*

*Proof sketch.* Consider the $\mathsf{PTL}_{\mathsf{P,Y}} \times \mathsf{K}$ formula $\varphi = \mathsf{Y}c_1$, which is satisfied by all pointed temporal graphs $(TG, (v, t_n))$ where $n \geq 2$ and the node $v$ was of colour $c_1$ at timestamp $t_{n-1}$. Interestingly, there is no $\mathcal{T}_{\text{glob}}[\mathcal{M}, \mathcal{Q}, \circ]$ for any $\mathcal{M}$, $\mathcal{Q}$, and $\circ$ that can express this property. The reason is as follows: the notion of temporal neighbourhood utilised by global TGNNs does not give a node $v$ access to its own past information. It can only access past informations of neighbours. Since this is the only form of temporal information used by such TGNNs, it excludes the ability to recognise whether $v$ itself was coloured $c_1$ in the past. For the full proof, see Appendix B.5. $\qquad\square$

Next, we turn to identifying a fragment of $\mathsf{PTL}_{\mathsf{P,Y}} \times \mathsf{K}$ captured by $\mathcal{T}_{\text{glob}}$. The architecture of global TGNNs appears to necessitate a class of MPNN components that effectively utilise temporal information. Specifically, we use MPNNs incorporating a learnable message function $\mathsf{msg}$ in their aggregation. This enables global TGNNs to process messages based on temporal information prior to aggregation. Thus, we use $\hat{\mathcal{M}}_{\mathsf{msg}}$ (see Section 4 to recall the definition of such GNNs). Concerning $\phi$ and $\circ$, meaning the operations by which temporal information is processed and then combined with static neighbourhood information during aggregation, we employ time2vec [22] and concatenation $\|$ of vectors, akin to models such as TGAT or Temporal Graph Sum [35]. Let $\mathcal{Q}_{\mathsf{time2vec}}$ be the class of all time2vec functions (see Appendix A.3 for a formal definition). We let $\mathcal{L}_2$ be the fragment of $\mathsf{PTL}_{\mathsf{P,Y}} \times \mathsf{K}$, whose formulas $\varphi$ are such that for all $Q\psi \in sub(\varphi)$ with $Q \in \{\mathsf{Y}, \mathsf{P}\}$ we have $\psi = \Diamond\chi$ for some formula $\chi$. For example $\mathsf{P}\Diamond c_1$ is a formula of $\mathcal{L}_2$, but $\mathsf{P}c_1$ is not.

**Theorem 11.** $\mathcal{L}_2 \leq \mathcal{T}_{\text{glob}}[\hat{\mathcal{M}}_{\mathsf{msg}}, \mathcal{Q}_{\mathsf{time2vec}}, \|]$.

*Proof sketch.* We use the inductive approach from Theorem 6, but now use a global TGNN $T_\varphi = (M, \phi, \mathsf{out})$. For each subformula $\varphi_i$, a layer $l^{(i)}$ in $M$ computes its semantics. Boolean subformulas are handled as in previous results and temporal subformulas $\mathsf{Y}\psi$ and $\mathsf{P}\psi$ are deferred, since their semantics are captured indirectly through $\Diamond$ subformulas, as ensured by the definition of $\mathcal{L}_2$. The key mechanism to do this is the interaction between $\phi$ and $\mathsf{msg}$: for each $Q\Diamond\psi$ with $Q \in \{\mathsf{P}, \mathsf{Y}\}$, $\mathsf{msg}$ selectively passes information based on $\phi(t)$, enforcing that only the correct previous or past timepoints contribute. If $\psi$ is of the form $\mathsf{Y}\Diamond\chi$ or $\mathsf{P}\Diamond\chi$, $\mathsf{msg}$ filters messages from exactly the previous timepoint or from all earlier timepoints, respectively. Otherwise, $\mathsf{msg}$ restricts aggregation to the current timepoint. Correctness follows by induction over time and subformula structure. A full proof is provided in Appendix B.6. $\qquad\square$

While employing time2vec functions is motivated by common practice, it is not essential for achieving the previous result. In fact, it suffices to have a function $\phi$ capable of mapping the values $0$, $-1$, and $t$, where $t \leq -2$, to distinct values. Similarly, $\|$ is not strictly necessary. The same result can be achieved, with a slightly adapted construction, using entrywise multiplication or addition as $\circ$.

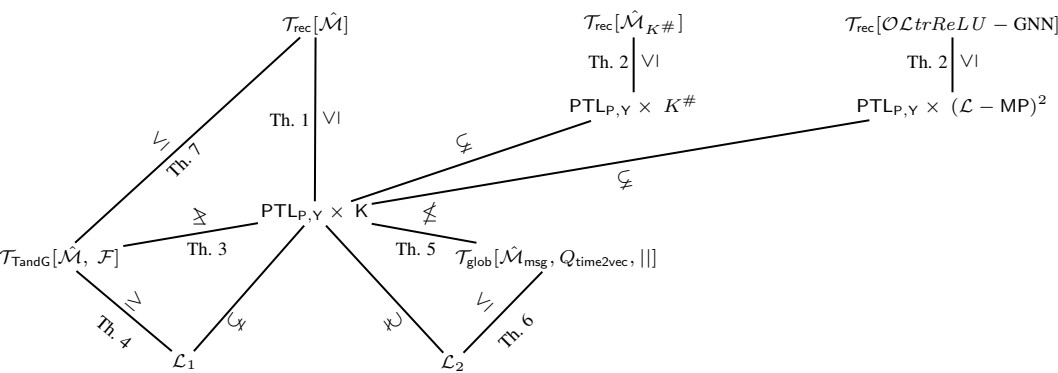

Figure 3: An overview of our expressive power results

Now, combining the previous results, namely Theorems 6 to 11, and the insight that $\Diamond(c_1 \wedge \mathsf{Y}\Diamond c_2) \in \mathcal{L}_2$ and $\mathsf{Y}c_1 \in \mathcal{L}_1$ we immediately get the following inexpressiveness results. Let $\mathcal{T}_1$ and $\mathcal{T}_2$ be two classes of TGNNs. We let $\mathcal{T}_1 \leq \mathcal{T}_2$ if for all $T \in \mathcal{T}_1$ there is $T' \in \mathcal{T}_2$ such that for all discrete, pointed temporal graphs $(TG, v)$ we have that $T(TG, v) = 1$ if and only if $T'(TG, v) = 1$. Accordingly, we define $\mathcal{T}_1 \not\leq \mathcal{T}_2$ if $\mathcal{T}_1 \leq \mathcal{T}_2$ does not hold, and we define $\mathcal{T}_1 \not\equiv \mathcal{T}_2$ if $\mathcal{T}_1 \not\leq \mathcal{T}_2$ or $\mathcal{T}_2 \not\leq \mathcal{T}_1$.

**Corollary 12.** $\mathcal{T}_{rec}[\hat{\mathcal{M}}] \not\leq \mathcal{T}_{TandG}[\hat{\mathcal{M}}, \mathcal{F}]$, $\mathcal{T}_{rec}[\hat{\mathcal{M}}] \not\leq \mathcal{T}_{glob}[\hat{\mathcal{M}}_{msg}, \mathcal{Q}_{time2vec}, ||]$, *and* $\mathcal{T}_{TandG}[\hat{\mathcal{M}}, \mathcal{F}] \not\equiv \mathcal{T}_{glob}[\hat{\mathcal{M}}_{msg}, \mathcal{Q}_{time2vec}, ||]$.

To complement the picture of expressiveness relationships obtained so far, we establish the following relationship between time-and-graph and recursive TGNNs; recall that $\mathcal{F} \subseteq \mathsf{FNN}[\mathsf{trReLU}]$ is the class of single layer FNNs with truncated-ReLU activations.

**Theorem 13.** $\mathcal{T}_{TandG}[\hat{M}, \mathcal{F}] \leq \mathcal{T}_{rec}[\hat{M}]$.

*Proof.* Let $T \in \mathcal{T}_{TandG}[\hat{M}, \mathcal{F}]$ with $T = (M_1, M_2, Cell, \mathsf{out})$, where each $M_i$ is of input dimensionality $m_i$ and output dimensionality $n_i$, and $Cell$ is of input dimensionality $n_1 + n_2$ and output dimensionality $n_3$. The construction of the witness $T' \in \mathcal{T}_{rec}[\hat{M}]$ is straightforward: $T'$ is given by $(M_3, \mathsf{out})$, where $M_3$ is given by $Cell \circ M_1 || M_2$, which represents the MPNN of input dimensionality $m_1 + m_2$ and output dimensionality $n_3$ simulating $M_1$ on the first $m_1$ input dimensions, $M_2$ on the last $m_2$ input dimensions, and then applies a message-passing layer, where the combination function ignores input from agg and uses $Cell$ otherwise. $\square$

Figure 3 summarises the expressive results we have established in the paper.

## 8 Conclusion

We have initiated the study of the expressive capabilities of TGNNs using logical languages, which is motivated by successful logical characterisation of static GNNs. As we have showed, product logics combining temporal and modal logics, are particularly well-suited to achieve this goal. In particular, we have studied three classes of TGNNs: recursive, time-and-graph, and global TGNNs. We have showed that recursive TGNNs can express all properties definable in the product logic $\mathsf{PTL}_{\mathsf{P,Y}} \times \mathsf{K}$, combining the standard temporal and modal logics. Moreover we have obtained analogous results by relating variants of $\mathsf{PTL}_{\mathsf{P,Y}} \times \mathsf{K}$ to recursive TGNNs that exploit specific classes of MPNNs. In contrast, neither time-and-graph or global TGNNs can express all properties definable in $\mathsf{PTL}_{\mathsf{P,Y}} \times \mathsf{K}$. The reason is that $\mathsf{PTL}_{\mathsf{P,Y}} \times \mathsf{K}$ allows for arbitrary interaction between logical operators expressing temporal and spatial properties. By restricting the form of these interaction, we have obtained fragments of $\mathsf{PTL}_{\mathsf{P,Y}} \times \mathsf{K}$ which can be captured by time-and-graph or global TGNNs, respectively.

Our results provide new insights into the expressive power of TGNNs and show that TGNN architectures significantly differ on the spatio-temporal properties they can capture. Better understanding of TGNN capabilities is crucial for choosing appropriate models for a downstream task and help in developing more powerful architectures. Since this is the first work on the logical expressiveness

of TGNNs, it has a number of interesting next steps that can be performed in future. Among others, we plan to investigate tight expressive bounds, and for a larger amount of TGNN architectures. Furthermore, we plan to transfer results known from research on product logics to TGNNs, including computational complexity analysis and finite model theory.

**Limitations.** The results established here are of a strictly formal nature. Our expressive results focus on showing which classes of TGNNs can express (i.e. detect) temporal graph properties expressible in particular logics. Hence our results do not aim to show which properties can be learnt in practice, but to show fundamental relations between expressiveness of TGNNs and product logics.

## Acknowledgments and Disclosure of Funding

This research is partially funded by the European Union (ERC, ExtenDD, project number: 101054714). Views and opinions expressed are however those of the authors only and do not necessarily reflect those of the European Union or the European Research Council. Neither the European Union nor the granting authority can be held responsible for them.

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

# A   Omitted definitions

In this section, we provide full formal definitions of concepts that were introduced informally in the main part of the paper.

## A.1   Feedforward neural networks

We present a formal definition of the classical feedforward neural network (FNN) model.

**Definition 14.** *An* FNN node $v$ *is a function from* $\mathbb{R}^m$ *to* $\mathbb{R}$, *where* $m \in \mathbb{N}$ *is the* input dimension, *computing*

$$v(x_1, \ldots, x_n) = \sigma(b + \sum_{i=1}^{n} w_i x_i),$$

*where* $w_i \in \mathbb{Q}$ *are the* weights, $b \in \mathbb{Q}$ *is the* bias, *and* $\sigma : \mathbb{R} \to \mathbb{R}$ *is the activation function. An* FNN layer $\ell$ *is a tuple consisting of some number* $n \in \mathbb{N}$ *FNN nodes* $(v_1, \ldots, v_n)$, *all having the same input dimensionality* $m$. *Then,* $\ell$ *computes the function from* $\mathbb{R}^m$ *to* $\mathbb{R}^n$, *where* $n$ *is the* output dimension, *given by*

$$\ell(x_1, \ldots, x_m) = (v_1(x_1, \ldots, x_m), \ldots, v_n(x_1, \ldots, x_m)).$$

*Finally, a* feedforward neural network (FNN) $N$ *is a tuple of some* $k \in \mathbb{N}$ *FNN layers* $(\ell_1, \ldots, \ell_k)$, *where the input dimension of* $\ell_{i+1}$ *is equal to the output dimension of* $\ell_i$ *for all* $i < k$. *Then,* $N$ *computes the function from* $\mathbb{R}^{m_1}$ *to* $\mathbb{R}^{n_k}$, *where* $m_1$ *is the input dimensionality of nodes in layer* $l_1$ *and* $n_k$ *is the number of nodes in layer* $l_k$, *given by*

$$N(x_1, \ldots, x_{m_1}) = \ell_k(\cdots \ell_1(x_1, \ldots, x_{m_1}) \cdots).$$

Given this definition, we denote by $\mathsf{FNN}[\mathsf{trReLU}]$ the class of all FNN where all nodes exclusively use $\mathsf{trReLU}(x) = \max(0, \min(x, 1))$ as the activation function.

## A.2   The product logic $\mathsf{PTL}_{\mathsf{P,Y}} \times \mathsf{K}$

In the following, we define one of the key logics we consider, namely $\mathsf{PTL}_{\mathsf{P,Y}} \times \mathsf{K}$.

**Definition 15.** *We define the logic* $\mathsf{PTL}_{\mathsf{P,Y}} \times \mathsf{K}$ *given by all formulae* $\varphi$ *defined by the grammar:*

$$\varphi ::= c_j \mid \neg\varphi \mid \varphi \wedge \varphi \mid \Diamond\varphi \mid \mathsf{Y}\varphi \mid \mathsf{P}\varphi$$

*where* $0 \le j < k$ *for some set of* $k$ *colours. Let* $(TG, v)$ *be a coloured, pointed temporal graph. Since the temporal operators in this logic are oblivious of the exact timestamps in* $TG$ *we assume that* $TG$ *is discrete. We say that* $(TG, v)$ *satisfies formula* $\varphi$, *written* $(TG, (v, t_n)) \models \varphi$, *if given:*

$$
\begin{aligned}
TG, (v, t_i) &\models c_j & \text{iff} & \quad (c_{t_i}(v))_j = 1, \\
TG, (v, t_i) &\models \neg\varphi & \text{iff} & \quad TG, (v, t_i) \not\models \varphi, \\
TG, (v, t_i) &\models \varphi_1 \wedge \varphi_2 & \text{iff} & \quad TG, (v, t_i) \models \varphi_1 \text{ and } TG, (v, t_i) \models \varphi_2, \\
TG, (v, t_i) &\models \Diamond\varphi & \text{iff} & \quad \text{there is } u \in V \text{ s.t. } \{v, u\} \in E_i \text{ and } TG, (u, t_i) \models \varphi, \\
TG, (v, t_i) &\models \mathsf{Y}\varphi & \text{iff} & \quad t \ne 0 \text{ and } TG, (v, t_i - 1) \models \varphi, \\
TG, (v, t_i) &\models \mathsf{P}\varphi & \text{iff} & \quad \text{there is } t_j < t_i \text{ such that } TG, (v, t_j) \models \varphi.
\end{aligned}
$$

## A.3   The product logics $\mathsf{PTL}_{\mathsf{P,Y}} \times (\mathcal{L}\text{-}\mathsf{MP}^2)$ and $\mathsf{PTL}_{\mathsf{P,Y}} \times K^{\#}$.

We give formal definitions of the logics $\mathsf{PTL}_{\mathsf{P,Y}} \times (\mathcal{L}\text{-}\mathsf{MP}^2)$ and $\mathsf{PTL}_{\mathsf{P,Y}} \times K^{\#}$ based on the modal logics presented in [5] and [32]. These follow the same line as the definition of $\mathsf{PTL}_{\mathsf{P,Y}} \times \mathsf{K}$ given in Definition 15.

**Definition 16.** *Formulae of* $\mathsf{PTL}_{\mathsf{P,Y}} \times (\mathcal{L}\text{-}\mathsf{MP}^2)$ *are defined by the grammar:*

$$\varphi ::= \top \mid c_i \mid \sum_{i=1}^{k} a_i \cdot \#\varphi \le b \mid \neg\varphi \mid \varphi \wedge \varphi \mid \mathsf{Y}\varphi \mid \mathsf{P}\varphi$$

*where $c_i$ ranges over propositional variables in a finite set $C$ of colours, and $a_i, b \in \mathbb{Q}$. Let $(TG, v)$ be a coloured, pointed temporal graph. Since the temporal operators in this logic are oblivious of the exact timestamps in $TG$ we assume that $TG$ is discrete. We say that $(TG, v)$ satisfies formula $\varphi$, written $(TG, (v, t_n)) \models \varphi$, if given:*

$$TG, (v, t_i) \models \top \qquad \textit{iff} \quad \textit{true,}$$

$$TG, (v, t_i) \models c_j \qquad \textit{iff} \quad (c_{t_i}(v))_j = 1,$$

$$TG, (v, t_i) \models \sum_{i=1}^{k} a_i \cdot \#\varphi_i \leq b \quad \textit{iff} \quad \sum_{i=1}^{k} a_i \cdot |\{u \in V \mid \{v, u\} \in E_t \text{ and } TG, (v, t_i) \models \varphi_i\}| \leq b,$$

$$TG, (v, t_i) \models \neg\varphi \qquad \textit{iff} \quad TG, (v, t_i) \not\models \varphi,$$

$$TG, (v, t_i) \models \varphi_1 \wedge \varphi_2 \qquad \textit{iff} \quad TG, (v, t_i) \models \varphi_1 \text{ and } TG, (v, t_i) \models \varphi_2,$$

$$TG, (v, t_i) \models \mathsf{Y}\varphi \qquad \textit{iff} \quad t_i \neq 0 \text{ and } TG, (v, t_i - 1) \models \varphi, \text{ and}$$

$$TG, (v, t_i) \models \mathsf{P}\varphi \qquad \textit{iff} \quad \text{there is } t' < t_i \text{ such that } TG, (v, t') \models \varphi.$$

Although it does not affect any of our results, we note that [5] consider MPNNs operating on directed graphs, whereas our focus is on static graphs with undirected edges. Consequently, there is a technical difference in the definition of the semantics of the $\sum_{i=1}^{k} a_i \cdot \#\varphi_i \leq b$ quantifier compared to [5].

**Definition 17.** *Formulae of $\mathsf{PTL}_{\mathsf{P,Y}} \times K^{\#}$ are defined by the grammar:*

$$\varphi ::= c \mid \sum_{i=1}^{k} a_i \cdot 1_\varphi + \sum_{i=1}^{k'} b_i \cdot \#\varphi \leq d \mid \neg\varphi \mid \varphi \wedge \varphi \mid \mathsf{Y}\varphi \mid \mathsf{P}\varphi$$

*where $c$ ranges over propositional variables in a finite set $C$ of colours, and $a_i, b_i, d \in \mathbb{Z}$.*

*Let $(TG, v)$ be a coloured, pointed temporal graph. Since the temporal operators in this logic are oblivious of the exact timestamps in $TG$ we assume that $TG$ is discrete. We say that $(TG, v)$ satisfies formula $\varphi$, written $(TG, (v, t_n)) \models \varphi$, if given:*

$$TG, (v, t_i) \models c_j \qquad \textit{iff} \quad (c_{t_i}(v))_j = 1,$$

$$TG, (v, t_i) \models \neg\varphi \qquad \textit{iff} \quad TG, (v, t_i) \not\models \varphi,$$

$$TG, (v, t_i) \models \varphi_1 \wedge \varphi_2 \qquad \textit{iff} \quad TG, (v, t_i) \models \varphi_1 \text{ and } TG, (v, t_i) \models \varphi_2,$$

$$TG, (v, t_i) \models \mathsf{Y}\varphi \qquad \textit{iff} \quad t_i \neq 0 \text{ and } TG, (v, t_i - 1) \models \varphi,$$

$$TG, (v, t_i) \models \mathsf{P}\varphi \qquad \textit{iff} \quad \text{there is } t' < t_i \text{ such that } TG, (v, t') \models \varphi, \text{ and}$$

$$TG, (v, t_i) \models \sum_{i=1}^{k} a_i \cdot 1_{\varphi_i} + \sum_{i=1}^{k'} b_i \cdot \#\varphi_i' \leq d \quad \textit{iff}$$

$$\sum_{i=1}^{k} a_i \cdot \begin{cases} 1 & \textit{if } TG, (v, t_i) \models \varphi_i \\ 0 & \textit{otherwise,} \end{cases} + \sum_{i=1}^{k'} b_i \cdot |\{u \in V \mid \{v, u\} \in E_t \text{ and } TG, (v, t_i) \models \varphi_i'\}| \leq d,$$

We note here that we utilised a normal form of $K^{\#}$ (refer to Theorem 1 in [32]) to simplify the definition of $\mathsf{PTL}_{\mathsf{P,Y}} \times K^{\#}$. As the authors point out, it is straightforward to observe that each formula of the original $K^{\#}$ syntax can be efficiently transformed into this normal form.

**The time function time2vec**

In the following, we define time2vec functions [22].

**Definition 18.** *We define $\phi(t_i - t_l) = \mathsf{t2v}(t_i - t_l)$ as*

$$\mathsf{t2v}(t)_j = \begin{cases} w_j t + b_j & \textit{if } j = 0, \\ \sigma(w_j t + b_j) & \textit{otherwise,} \end{cases}$$

*where $w_j, b_j \in \mathbb{Q}$ and $\sigma$ is some periodic activation function.*

Correspondingly, we denote the class of all such functions by $\mathcal{Q}_{\mathsf{time2vec}}$.

## B    Omitted proofs

In this section, we provide all the formal proofs for the results presented in this work. This includes comprehensive proofs for the results of the paper that were merely outlined in the main section.

### B.1    Proof of Theorem 6

**Theorem 6.** $\mathsf{PTL_{P,Y}} \times \mathsf{K} \leq \mathcal{T}_{rec}[\hat{\mathcal{M}}]$.

*Proof.* Let $\varphi$ be a formula of $\mathsf{PTL_{P,Y}} \times \mathsf{K}$ as defined in Definition 5 with $m$ atomic subformulas. Let

$$\varphi_1, \ldots, \varphi_m, \varphi_{m+1}, \ldots, \varphi_n$$

be an enumeration of the subformulas of $\varphi$ such that all atomic formulas are the $\varphi_1, \ldots, \varphi_m$, and $\varphi_i \in sub(\varphi_j)$ implies $i \leq j$. In particular, we have $\varphi_n = \varphi$.

We begin by describing how the TGNN $T_\varphi$ is constructed. We have $T_\varphi = (M, \mathsf{out})$, where the MPNN $M = (l^1, \ldots, l^{n-m+1})$ and the layer $l^{(i)}$ with $i \leq n - m$ is given by $(\mathsf{comb}^{(i)}, \sum)$ with $\mathsf{comb}^{(i)}(\boldsymbol{x}, \boldsymbol{y}) = \mathsf{trReLU}(C\boldsymbol{x} + A\boldsymbol{y} + b)$, and where $\sum$ denotes entrywise sum as aggregation. The exact form of the $3n \times 3n$ matrices $C$, $A$, and the $n$-dimensional vector $b$ depends on $\varphi_{m+i}$ as follows:

- if $\varphi_{m+i} = \neg\varphi_j$, we have $C_{m+i,j} = -1$, and $b_{m+i} = 1$,

- if $\varphi_{m+i} = \varphi_{j_1} \wedge \varphi_{j_2}$, we have $C_{m+i,j_1} = C_{m+i,j_2} = 1$, and $b_{m+i} = -1$,

- if $\varphi_{m+i} = \Diamond\varphi_j$, we have $A_{m+i,j} = 1$,

- if $\varphi_{m+i} = \mathsf{Y}\varphi_j$, we have $C_{m+i,n+j} = 1$,

- if $\varphi_{m+i} = \mathsf{P}\varphi_j$, we have $C_{m+i,2n+j} = 1$, and

for all $j \leq m$ we have $C_{j,j} = 1$. All other entries of $C$, $A$, and $b$ are zero. This implies that all layers $l_i$ with $i \leq n - m$ use the same parameters. The layer $l^{n-m+1} = (\mathsf{comb}^{n-m+1}, \sum)$ is given by $\mathsf{comb}^{n-m+1}(\boldsymbol{x}, \boldsymbol{y}) = \mathsf{trReLU}(C'\boldsymbol{x} + A'\boldsymbol{y} + \boldsymbol{0})$, where $A'$ is the $2n \times 3n$ all-zero matrix, $\boldsymbol{0}$ is the $2n$-dimensional all-zero vector, and $C'$ is the $2n \times 3n$ matrix with $C'_{j,j} = 1$, $C'_{n+j,n+j} = 1$, and $C'_{n+j,2n+j} = 1$ for all $j \leq n$. All other entries are zero. The output function is given by $\mathsf{out}(x_1, \ldots, x_{2n}) = \mathsf{trReLU}(x_n)$. It is straightforward to see that $T_\varphi \in \mathcal{T}_{rec}[\mathcal{M}]$.

Let $(TG, v)$ be a pointed temporal graph where $TG$ is of length $k$ and $V$ is its set of nodes. Regarding correctness, we prove the following statement: for all nodes $u \in V$, all timepoints $t_i$ with $i \leq k$ of $TG$ and subformulas $\varphi_j$ with $j \leq n$ of $\varphi$ we have that

a) $TG, (u, t_i) \models \varphi_j$ if and only if $\mathbf{h}_u^{(\min(0,j-m))}(t_i)_j = 1$,

b) $TG, (u, t_i - 1) \models \varphi_j$ if and only if $\mathbf{h}_u^{(\min(0,j-m))}(t_i)_{n+j} = 1$, and

c) $TG, (u, t_{i'}) \models \varphi_j$ for some $i' < i$ if and only if $\mathbf{h}_u^{(\min(0,j-m))}(t_i)_{2n+j} = 1$.

We prove this statement via strong induction on $i$ and $j$.

**Case: timestamp $t_1$.**    First, let $i = 1$ and $j \in \{1, \ldots, m\}$, and fix some $u \in V$. The assumption $j \in \{1, \ldots, m\}$ implies that $\varphi_j$ is an atomic formula. Then, statement (a) is directly implied by the form of $h_u^{(0)}(t_1)$, including the colours of node $u$. Similarly, we have that statements (b) and (c) are given as $T_\varphi$ initialises the dimensions $n$ to $3n$ of $h_u^{(0)}(t_1)$ with 0, which is correct as we are considering the first timestamp of $TG$. Next, assume that the statement holds for $i = 1$, $j$, and all $u \in V$. Consider the case of $j + 1$, fix some $u \in V$, and focus on statement (a). The Boolean cases $\varphi_{j+1} = \neg\varphi_{l_1}$ and $\varphi_{j+1} = \varphi_{l_1} \wedge \varphi_{l_2}$ are a straightforward implication of the form of matrix $C$, vector $b$, as well as the activation function $\mathsf{trReLU}$, and the fact that statement (a) holds for all $l_1, l_2 \leq j$. Similarly, the case $\Diamond\varphi_{l_1}$ is implied by the form of matrix $A$, the activation function

trReLU, and that statement (a) holds for all $l_1 \leq j$ and all $w \in V$, including the neighbours of $u$. In the case of $\varphi_{j+1} = \mathsf{Y}\varphi_{l_1}$ or $\varphi_{j+1} = \mathsf{P}\varphi_{l_1}$, we rely on the fact that $i = 1$, which means that these are necessarily false. By the hypothesis, this is implied by (b) and (c) for all $l_1 \leq j$ as matrix $C$ utilises the corresponding dimensions $n + l_1$ and $2n + l_1$, respectively, which are all 0. The fact that we consider $i = 1$ and that all dimenstions $l > n$ are 0 also immediately implies statements (b) and (c).

**Case: timestamp $t_i$ with $i > 1$.** Next, assume that statements (a) to (c) hold for $i$, all $j \leq n$, and all $u \in V$, and consider the case $i + 1$. Once again, the argument for statement (a) with $j \in \{1, \ldots, m\}$ is directly implied by the fact that $T_\varphi$ stores the colours of each node $u$ in the respective dimension. Before addressing (b) and (c) for these $j$, consider the following two observations:

1. TGNN $T_\varphi$ ensures for all timestamps $t_i$, $j \leq n$, and $u \in V$ that $\mathbf{h}_u^{(0)}(t_{i+1})_{n+j} = \mathbf{h}_u^{(n-m)}(t_i)_j$, and it ensures that $\mathbf{h}_u^{(0)}(t_{i+1})_{2n+j} = 1$ if and only if $\mathbf{h}_u^{(n-m)}(t_{i'})_j$ for some $i' \leq i$.

2. TGNN $T_\varphi$ ensures for all timestamps $t_i$, $u \in V$ that $\mathbf{h}_u^{(0)}(t_i)_j = \mathbf{h}_u^l(t_i)_j$ holds for all $j \in \{n+1, \ldots, 3n\}$ and $l \leq n - m$, and it ensures that $\mathbf{h}_u^{(j)}(t_i)_j = \mathbf{h}_u^l(t_i)_j$ holds for all $j \leq n$ and $j \leq l \leq n - m$.

Informally, the first observation implies that dimensions $n + 1$ to $2n$ store the previous state of a node, and $2n + 1$ to $3n$ store the disjunction over all previous states of a node, where 1 is interpreted as true and 0 as false. This is achieved by the way we built layer $l^{n-m+1}$. The second observation simply states that, within a single timepoint $t_i$, the TGNN $T_\varphi$ does not alter dimensions $n + 1$ to $3n$ within layers $l^1$ to $l^{n-m}$ and that if the semantics of $\varphi_j$ are computed in the $j$-th layer, they are preserved in subsequent layers. Now, consider statements (b) and (c) for $i + 1$, $j \in \{1, \ldots, m\}$ and some fixed $u \in V$. Here, the first observation and induction hypothesis directly imply these statements. Therefore, keep $u \in V$ fixed and consider $j + 1$, while assuming that (a) to (c) hold for all $j' \leq j$. The Boolean and modal cases are argued exactly as before. Thus, consider $\varphi_{j+1} = \mathsf{Y}\varphi_l$ for some $l \leq j$. Using the first and second observations, we know that the semantics of $\varphi_l$ at timepoint $t_i$ are stored in $\mathbf{h}_u^{(j+1)}(t_{i+1})_{n+l}$, which is utilised by $T_\varphi$ to compute the semantics of $\mathsf{Y}\varphi_l$. Given the induction hypothesis, this is correct. Similarly, the case $\varphi_{j+1} = \mathsf{P}\varphi_l$ is implied by the two observations and the induction hypothesis, which state that $\mathbf{h}_u^{(j+1)}(t_{i+1})_{2n+l}$ stores a 1 if $\varphi_l$ was true at any timepoint before $t_{i+1}$ and 0 otherwise. Given this, the correctness is immediate.

Finally, the correctness of the theorem is given by statement (a) for $t_k$ and $\varphi_n = \varphi$ in combination with observation that out uses the $n$-th dimension of $\mathbf{h}_v^{n-m+1}(t_k)$ to compute the overall output. $\quad\square$

## B.2 Proof of Theorem 7

For a formal definition of the logics $\mathsf{PTL}_{\mathsf{P,Y}} \times (\mathcal{L}\text{-}\mathsf{MP}^2)$ and $\mathsf{PTL}_{\mathsf{P,Y}} \times K^{\#}$, we refer to the corresponding subsection of Appendix A.3.

The key results we rely on are Theorem 26 of [5] and Theorem 1 of [32]. However, we need a stronger form of these results that includes the capturing of the semantics of each subformula with a MPNN.

**Definition 19.** *Let $\varphi$ be an inductively built formula interpreted over pointed static graphs, and let $M$ be an MPNN. We say that $M$ inductively captures $\varphi$ if there exists an enumeration of all subformulas $\varphi_1, \ldots, \varphi_m$ of $\varphi$, where $\varphi_i \in sub(\varphi_j)$ implies $i \leq j$, such that for all pointed graphs $(G, v)$ and formulas $\varphi_i$ there is layer $j_i$ of $M$ such that*

- *if $(G, v) \models \varphi_i$, then $\mathbf{h}^l(v)_i = 1$ for all $l \geq j_i$, and*

- *if $(G, v) \not\models \varphi_i$, then $\mathbf{h}^l(v)_i = 0$ for all $l \geq j_i$,*

*where $\mathbf{h}^l(v)$ represents the state of $v$ computed by the $j_i$-th layer of $M$. Furthermore, we require that $j_{i-1} \leq j_i$ for all $i \in \{1, \ldots, m\}$. We extend the notion of inductive capture to a logic $\mathcal{L}$ over pointed static graphs and a class of MPNNs $\mathcal{M}$ in the obvious way and denote it by $\mathcal{L} \trianglelefteq \mathcal{M}$.*

Given this understanding, a close examination of the arguments employed in [5] and [32] directly implies the inductive capture of the logics $\mathcal{L}\text{-}\mathsf{MP}^2$ and $K^{\#}$ by the respective classes of MPNNs.

**Lemma 1.** *Let $\mathcal{L}$-MP$^2$ and $K^\#$ be the logics introduced in [5] and [32] and $\mathcal{OL}$trReLU-MPNN and $\mathcal{M}_{K\#}$ the respective classes of MPNNs. We have that $\mathcal{L}$-MP$^2 \trianglelefteq \mathcal{OL}$trReLU-MPNN and $K^\# \trianglelefteq \mathcal{M}_{K\#}$.*

*Proof.* Both results, namely Theorem 26 from [5] and Theorem 1 from [32], are constructive in nature: for each formula $\varphi$ the authors construct an MPNN $M_\varphi$ that captures its semantics. Moreover, these constructions are inductive, meaning that all subformulas of $\varphi$ are enumerated and the MPNN $M_\varphi$ is built such that it captures the semantics of the $i$-th subformula with its $i$-th layer and preserves it in subsequent layers. This establishes that these results in fact imply inductive capturing. $\square$

Now, we are set to prove that both $\mathsf{PTL}_{\mathsf{P},\mathsf{Y}} \times (\mathcal{L}\text{-}\mathsf{MP}^2)$ and $\mathsf{PTL}_{\mathsf{P},\mathsf{Y}} \times K^\#$ are captured by the respective classes of TGNNs.

**Theorem 7.** $\mathsf{PTL}_{\mathsf{P},\mathsf{Y}} \times K^\# \leq \mathcal{T}_{rec}[\mathcal{M}_{K\#}]$ *and* $\mathsf{PTL}_{\mathsf{P},\mathsf{Y}} \times (\mathcal{L}\text{-}\mathsf{MP}^2) \leq \mathcal{T}_{rec}[\mathcal{OL}\text{trReLU-}GNN]$.

*Proof.* First, we address the statement that $\mathsf{PTL}_{\mathsf{P},\mathsf{Y}} \times (\mathcal{L}\text{-}\mathsf{MP}^2) \leq \mathcal{T}_{rec}[\mathcal{OL}\text{trReLU-GNN}]$. Let $\varphi \in \mathsf{PTL}_{\mathsf{P},\mathsf{Y}} \times (\mathcal{L}\text{-}\mathsf{MP}^2)$ with $k$ different subformulas of the form $\mathsf{Y}\psi$ or $\mathsf{P}\psi$. Let

$$\varphi_1, \ldots, \varphi_{m_1}, \varphi_{m_1+1}, \ldots, \varphi_{m_2}, \varphi_{m_2+1}, \ldots, \varphi_{m_k}, \varphi_{m_k+1}, \ldots, \varphi_n$$

be an enumeration of the subformulas of $\varphi$ such that $\varphi_i \in sub(\varphi_j)$ implies $i \leq j$, $\varphi_i \neq \mathsf{Y}\psi$ and $\varphi_i \neq \mathsf{P}\psi$ if $i \notin \{m_1, \ldots, m_k\}$, and we have $\varphi_n = \varphi$. Furthermore, we assume for each set $S_i = \{\varphi_{m_i+1}, \ldots, \varphi_{m_{i+1}-1}\}$ with $m_0 = 1$ that the enumeration is such that Lemma 1 applies to each formula of $S_i$, where we interpret potential occurrence of $\varphi_j$ with $j \leq m_i$ as some fresh atomic formula.

Let $T_\varphi = (M, \mathsf{out})$, where $\mathsf{out}(x_1, \ldots, x_{2n}) = \mathsf{trReLU}(x_n)$ and $M$ is constructed as follows. The initial layers are constructed to capture the semantics of the formulas $\varphi_1$ to $\varphi_{m_1-1}$ in dimensions 1 to $m_1 - 1$. The existence of such message-passing layers is guaranteed by the existence of MPNNs capturing the formulas of $S_0 = \{\varphi_1, \ldots, \varphi_{m_1-1}\}$, as indicated by Lemma 1. Furthermore, these message-passing layers are built with input and output dimensionality of $3n$, where dimensions $m_1$ to $3n$ are mapped by the identity function within the range $[0, 1]$. The argument that these can be interconnected in $M$ to form a well-formed MPNNs is given by the fact that the class $\mathcal{OL}$trReLU-GNN encompasses allows for arbitrary single-layer FNN with truncated-ReLU activations as combinations. Next, consider $\varphi_{m_1}$ which is either $\mathsf{Y}\varphi_i$ or $\mathsf{P}\varphi_i$ for some $i \leq m_1$. In the case of $\mathsf{Y}\varphi_i$, we add a layer that maps dimension $n + i$ to dimension $m_1$, and in the case of $\mathsf{P}\varphi_i$, we add a layer that maps dimension $2n + i$ to dimension $m_1$. Again, this is feasible due to the fact that $\mathcal{OL}$trReLU-MPNN includes arbitrary single-layer FNN with truncated-ReLU activations as combinations. Subformulas $\varphi_{m_1+1}$ to $\varphi_{m_2-1}$ are handled like $\varphi_1$ to $\varphi_{m_1-1}$ using the stack of MPNNs implied by Lemma 1 for $S_1 = \{\varphi_{m_1+1}, \ldots, \varphi_{m_2-1}\}$, combined accordingly. Then, $\varphi_{m_2}$ is addressed like $\varphi_{m_1}$ and so forth. Finally, we add a layer $l$ that computes the exact same function as $l^{n-m+1}$ in the proof of Theorem 6, which ensures that previous and past semantics are preserved. Given this construction, we have that $T_\varphi \in \mathcal{T}_{rec}[\mathcal{OL}\text{trReLU-MPNN}]$.

The correctness argument follows the exact same line of reasoning as that in Theorem 6, namely, arguing inductively over the timepoints $i$ of a temporal graph and subformulae $\varphi_j$. However, because we have not explicitly constructed $T_\varphi$, we utilize the following arguments. For non-temporal subformulae $\varphi_j$, we rely on the fact that Lemma 1 applies to $\mathcal{OL}$trReLU-MPNN, which means that $\varphi_j$ is captured after the respective layer. Note that this includes the observation that the assumptions we made for $S_i$ in order to apply Lemma 1 makes no difference in the overall computation of $M$. Otherwise, we use precisely the same arguments.

Finally, for the case of $\mathsf{PTL}_{\mathsf{P},\mathsf{Y}} \times K^\# \leq \mathcal{T}_{rec}[\mathsf{PTL}_{\mathsf{P},\mathsf{Y}} \times K^\#]$, we note that Lemma 1 is applicable to both $K^\#$ and $\mathcal{M}_{K\#}$. Moreover, the properties regarding the combination of MPNN layers we needed above are also given for $\mathcal{M}_{K\#}$. Therefore, the argumentation presented above is equivalently applicable. $\square$

## B.3 Proof of Theorem 8

**Theorem 8.** $\mathsf{PTL}_{\mathsf{P},\mathsf{Y}} \times \mathsf{K} \not\leq \mathcal{T}_{TandG}[\mathcal{M}, \mathcal{C}]$, *for all $\mathcal{M}$ and $\mathcal{C}$.*

*Proof.* Let $\varphi \in \mathsf{PTL}_{\mathsf{P},\mathsf{Y}} \times \mathsf{K}$ be the formula $\varphi = \Diamond(c_1 \wedge \mathsf{Y}\Diamond c_2)$. We prove that there is no TGNN $T \in \mathcal{T}_{\mathsf{TandG}}[\mathcal{M}, \mathcal{C}]$ for any class of MPNNs $\mathcal{M}$ and functions $\mathcal{C}$ such that for all pointed temporal graphs $(TG, v)$ we have that $(TG, v) \models \varphi$ if and only if $T(TG, v) = 1$.

We consider the two pointed temporal graphs $(TG, v)$ and $(TG', v')$ fully specified in form of Figure 2. It can be easily verified that $(TG, v) \models \varphi$ and $(TG', v') \not\models \varphi$. Let $T \in \mathcal{T}_{\mathsf{TandG}}[\hat{\mathcal{M}}, \mathcal{C}]$ be some TGNN. It is straightforward to see that for $M_1$ of $T$, we have that (a) $M_1(G_1, v) = M_1(G'_1, v')$, $M_1(G_1, u_1) = M_1(G'_1, w'_1)$, $M_1(G_1, u_2) = M_1(G'_1, w'_2)$, $M_1(G_1, w_1) = M_1(G'_1, u'_1)$, and $M_1(G_1, w_2) = M_1(G'_1, u'_2)$. This follows from the fact that $M_1$ is a function and the input of the aggregation is mutliset without any ordering. Similarly, this implies (b) $M_1(G_2, x) = M_1(G'_2, x')$ for $x \in \{v, u_1, u_2, w_1, w_2\}$. Also, we have that $M_2$ outputs the same vector for each pair of nodes $x$ and $x'$ at timestamp $t_1$, due to the fact that all labels are assumed to be $\mathbf{0}$ and $E_1 = E'_1$. Thus, the equalities of (a) are preserved by the application of $Cell$ in the first timestamp $t_1$.

Now, using (a), the same kind of argument gives that $M_2$ outputs the same vector for $v$ and $v'$ at timestamp $t_2$. Then, combining this with (b), we find that the function $Cell$ receives identical inputs in the cases of $(TG, v)$ and $(TG', v')$ at timestamp $t_2$, leading to the conclusion that either both are accepted or both are rejected by $T$. Thus, $T$ does not accept the exact same set of pointed temporal graphs as $\varphi$. $\qquad\square$

## B.4 Proof of Theorem 9

**Theorem 9.** $\mathcal{L}_1 \leq \mathcal{T}_{\mathsf{TandG}}[\hat{\mathcal{M}}, \mathcal{F}]$.

*Proof.* The proof follows a similar line of reasoning as our previous results, notably the arguments presented in the proof of Theorem 6. Let $\varphi$ be a formula of the fragment $\mathcal{L}_1$, and let $\varphi_1, \ldots, \varphi_n$ where $\varphi_i \in sub(\varphi_j)$ implies $i \leq j$. Specifically, we have $\varphi_n = \varphi$.

Let $T_\varphi = (M_1, M_2, Cell, \mathsf{out})$, where $\mathsf{out}(x_1, \ldots, x_{2n}) = \mathsf{trReLU}(x_n)$. The components $M_1$, $M_2$, and $Cell$ are constructed as outlined below. $M_1$ and $M_2$ consist of layers $l_1^{(i)}$ and $l_2^{(i)}$, respectively. In the case of $M_1$ we include a layer $l_1^{(0)}$ that maps vector $\boldsymbol{x} \in \{0, 1\}^m$ to $\boldsymbol{x} \,\|\, \mathbf{0} \in \{0, 1\}^m \times \{0\}^{n-m}$ and in the case of $M_2$ we include a layer $l_2^{(0)}$ that maps vector $\boldsymbol{x} \in \{0, 1\}^{2n}$ to $\mathbf{0} \,\|\, \boldsymbol{x} \in \{0\}^n \times \{0, 1\}^{2n}$. Each layer $l_j^{(i)}$ with $j \in \{1, 2\}$ is represented by $(\mathsf{comb}_j^{(i)}, \sum)$, where $\sum$ means entrywise sum as aggregation and $\mathsf{comb}_j^{(i)}(\boldsymbol{x}, \boldsymbol{y}) = \mathsf{trReLU}(C_j \boldsymbol{x} + A_j \boldsymbol{y} + \boldsymbol{b}_j)$, where $C_1, A_1 \in \{0, 1\}^{n \times n}$, $\boldsymbol{b}_1 \in \{0, 1\}^n$ and $C_2, A_2 \in \{0, 1\}^{3n \times 3n}$, $\boldsymbol{b}_2 \in \{0, 1\}^{3n}$. The function $Cell$ is represented by an FNN $N_{Cell}$ of input dimensionality $3n$ and output dimensionality $2n$. The entries of $C_j$, $A_j$, $\boldsymbol{b}_j$ with $j \in \{1, 2\}$, and the exact form of $N_{Cell}$ are determined by the enumeration of the subformulas $\varphi_i$. We distinguish three cases, depending on the nature of $\varphi_i$.

Firstly, let $\varphi_i$ be such that there is no subformula $Q\psi \in sub(\varphi_i)$ with $Q \in \{\mathsf{Y}, \mathsf{P}\}$. In this case, we set the parameters of $C_1$, $A_1$, and $\boldsymbol{b}_1$ based on $\varphi_i$ as shown in the proof of Theorem 6. Informally, this means the semantics of $\varphi_i$ are checked by $M_1$.

Secondly, consider $\varphi_i$ where for every subformula $c \in sub(\varphi_i)$, there exists $Q\psi \in sub(\varphi_i)$ with $Q \in \{\mathsf{Y}, \mathsf{P}\}$ such that $c \in sub(\psi)$. We configure the parameters $C_2$, $A_2$, and $\boldsymbol{b}_2$ based on $\varphi_i$ following the procedure described in the proof of Theorem 6. However, for $\varphi_i = \mathsf{Y}\varphi_j$, we use input dimension $n + j$, and for $\varphi_i = \mathsf{P}\varphi_j$, we use $2n + j$. We also add a final layer to $M_2$, which maps vectors $\boldsymbol{x}_1 \,\|\, \boldsymbol{x}_2 \,\|\, \boldsymbol{x}_3$, where $\boldsymbol{x}_j \in \{0, 1\}^n$, to $\boldsymbol{x}_1 \,\|\, \boldsymbol{x}_3$.

Thirdly, formulas $\varphi_i$ that do not belong to the first or second category are handled by $N_{Cell}$ of input dimensionality $3n$ and output dimensionality $2n$. The definition of the fragment $\mathcal{L}_1$ ensures that $\varphi_i = \neg\psi$ or $\varphi_i = \psi_1 \wedge \psi_2$. We refer to works such as [36], which demonstrate how to construct single-layer FNNs to check Boolean conditions. Besides processing these $\varphi_i$, the FNN $N_{Cell}$ maps input $x_i$ corresponding to $\varphi_i$ of the first category by using the identity (simply realised by $\mathsf{trReLU}(x)$) to output $y_i$ and input $x_{n+i}$ corresponding to $\varphi_i$ from the second category are mapped identically to output $y_i$ as well. Additionally, for each $j \leq n$, we add a component that computes $\mathsf{trReLU}(x_j + x_{n+j} + x_{2n+j})$ as the $n + j$th output, ensuring that all $\mathsf{P}\psi$ subformulas are correctly processed.

Regarding correctness, the argument follows the same inductive approach as in Theorem 6. To avoid repetition, we provide a high-level outline here. Concerning the MPNN $M_1$, it is immediately implied by the construction of Theorem 6 that it functions as expected, meaning that it correctly captures the semantics of $\varphi_i$ of the first category. This is due to the fact that it is only utilised to check formulas devoid of temporal operators. For $M_2$, consider that at the initial timestamp $t_1$, its inputs are pointed graphs $((V_1, E_1, [u \mapsto \mathbf{0}]), v)$, where all labels of $G_1$ are replaced by $\mathbf{0}$. Since $M_2$ is only used to verify formulas concerning past timepoints, the base case is correct. Otherwise, the inputs for $M_2$ are $((V_i, E_i, [u \mapsto \mathbf{h}_u(t_{i-1})]), v)$, where $\mathbf{h}_u(t_{i-1})$ denotes the output of $Cell$ for node $u$ at the previous timestamp. Here, our construction ensures that $\mathbf{h}_u(t_{i-1}) = \mathbf{h}_u^{\mathsf{Y}}(t_{i-1}) \,\|\, \mathbf{h}_u^{\mathsf{P}}(t_{i-1})$, where $\mathbf{h}_u^{\mathsf{Y}}(t_{i-1}) \in \{0,1\}^n$ contains the semantics of each subformula $\varphi_i$ at timestamp $t_{i-1}$ and $\mathbf{h}_u^{\mathsf{P}}(t_{i-1}) \in \{0,1\}^n$ contains semantics of each subformula $\varphi_i$, disjunctively combined over all $t_j$ with $j \leq i-1$. Due to the way $\mathcal{L}_1$ is defined, these are the necessary informations to compute the semantics of $\varphi_i$ of the second category at timepoint $t_i$. Finally, $N_{Cell}$ uses the outputs of $M_1$ and $M_2$ to compute the semantics of the remaining subformulas. Due to the way the fragment $\mathcal{L}_1$ is defined, these can not be of the form $\Diamond\psi$, which $N_{Cell}$ could not handle, or $Q\psi$ with $Q \in \{\mathsf{Y}, \mathsf{P}\}$, which are already handled by $M_2$. This leaves only Boolean formulas. Otherwise, $N_{Cell}$ ensures consistency and, thus, produces the output $\mathbf{h}_u(t_i) \,\|\, \mathbf{h}_u^{\mathsf{P}}(t_i)$, where $\mathbf{h}_u(t_i) \in \{0,1\}^n$ contains the semantics of all subformulas of $\varphi$ at timepoint $t_i$ and $\mathbf{h}_u^{\mathsf{P}}(t_i) \in \{0,1\}^n$ contains semantics of all subformulas, disjunctively combined over all $t_j$ with $j \leq i$. Finally, the output function $\mathsf{out}(x_1, \ldots, x_{2n}) = \mathsf{trReLU}(x_n)$ ensures that $T_\varphi$ outputs the semantics of $\varphi_n = \varphi$. $\qquad\square$

### B.5   Proof of Theorem 10

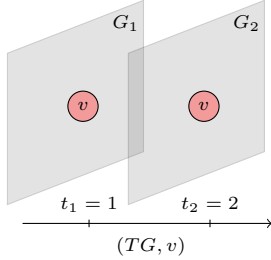 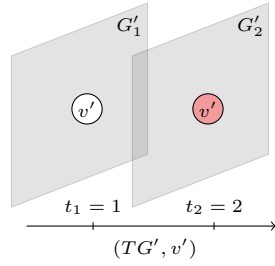

Figure 4: Pointed temporal graphs $(TG, v)$ and $(TG', v')$, both including two snapshots, used as an counterexample in the proof of Theorem 10. Here, colour $c_1$ is denoted by a red filling (applies for node $v$ in $G_1$, $G_2$ and node $v'$ in $G_2'$).

**Theorem 10.** $\mathsf{PTL}_{\mathsf{P},\mathsf{Y}} \times \mathsf{K} \not\leq \mathcal{T}_{glob}[\mathcal{M}, \mathcal{Q}, \circ]$, *for all $\mathcal{M}$, $\mathcal{Q}$, and $\circ$.*

*Proof.* Let $\varphi = \mathsf{Y}c_1$. It is evident that $\varphi$ is satisfied by all pointed temporal graphs $(TG, v)$ of length $n \geq 2$ such that $v$ was of colour $c_1$ at timestamp $t_1$.

Consider the two pointed temporal graphs $(TG, v)$ and $(TG', v')$, as specified by Figure 4. It is clear that $(TG, v) \models \varphi$ and $(TG', v') \not\models \varphi$. Now, let $T \in \mathcal{T}_{glob}[\mathcal{M}, \mathcal{Q}, \circ]$ for some $\mathcal{M}$, $\mathcal{Q}$, and $\circ$. The argument is simple: We have $h_v^0(t_2) = h_{v'}^{(0)}(t_2)$ in the respective computation of $T(TG, v)$ and $T(TG', v')$, and the input to $\mathsf{agg}$ is the empty set in both cases, meaning that its output is $\mathbf{0}$ in both cases. Thus, as $T$ either accepts both or none of the temporal graphs $(TG, v)$ and $(TG', v')$, meaning that it does not accept the exact set of temporal graphs that satisfy $\varphi$. $\qquad\square$

### B.6   Proof of Theorem 11

In the following result, we utilise time2vec functions [22] in the constructed TGNN. A formal definition can be found in Appendix A. We remark that in the following result we exclusively utilise the 0-th element of t2v functions $\phi$, indicating that the result is independent of the exact form of activation $\sigma$ used in these functions.

**Theorem 11.** $\mathcal{L}_2 \leq \mathcal{T}_{glob}[\hat{\mathcal{M}}_{\mathsf{msg}}, \mathcal{Q}_{\mathsf{time2vec}}, \|\,]$.

*Proof.* Let $\varphi \in \mathcal{L}_2$ and let

$$\varphi_1, \ldots, \varphi_m, \varphi_{m+1}, \ldots, \varphi_n$$

be an enumeration of the subformulas of $\varphi$ such that all atomic formulas are the $\varphi_1, \ldots, \varphi_m$, and $\varphi_i \in sub(\varphi_j)$ implies $i \leq j$. In particular, we consider an enumeration such that $\varphi_n = \varphi$.

The global (in time) TGNN $T_\varphi = (M, \phi, \text{out}) \in \mathcal{T}_{\text{glob}}[\hat{\mathcal{M}}_{\text{msg}}, \mathcal{Q}_{\text{time2vec}}, ||\,]$, where $\phi$ is some t2v function with $w_0 = 1$, $b_0 = 0$, meaning the 0-th element is the identity, and $\text{out}(x_1, \ldots, x_{2n}) = \text{trReLU}(x_n)$. We note that $M \in \hat{\mathcal{M}}_{\text{msg}}$ which means that aggregation is given by $\sum \text{msg}(x)$, where msg is realised by some one layer FNN $N_{\text{msg}}$ with truncated-ReLU activation. The MPNN $M$ is build such that for each $\varphi_i$ there is a layer $l^{(i)}$ in the same manner as done in Theorem 6. However, each layer has input dimension $n$ and output dimension $n$, which stands in contrast to previous constructions. This is due to the fact that, the semantics at previous or past timestamps are respected via aggregation. This works as follows.

Assume that $\varphi_i$ is Boolean. Then, it is handled in layer $l^{(i)}$ exactly as shown in the proof of Theorem 6. In the case of $\varphi_i = Q\psi$ with $Q \in \{\mathsf{Y}, \mathsf{P}\}$, due to the way $\mathcal{L}_2$ is defined, these subformulas must occur in form of $Q\Diamond\chi$. For modal subformulas $\Diamond$, we therefore distinguish three cases, namely that $\mathsf{Y}\Diamond\varphi_j$, $\mathsf{P}\Diamond\varphi_j$, or something else. In the case that $\mathsf{Y}\Diamond\varphi_j$, we build msg, represented by $N_{\text{msg}}$, such that it maps dimension $j$ identically if $\phi(t)_0 = -1$; otherwise, it maps dimension $j$ to 0. In the case $\mathsf{P}\Diamond\varphi_j$, we build $N_{\text{msg}}$ such that it maps dimension $j$ identically if $\phi(t)_0 \leq -1$; otherwise, it maps dimension $j$ to 0. In the third case, we build $N_{\text{msg}}$ such that it maps dimension $j$ identically if $\phi(t)_0 = 0$; otherwise, it maps dimension $j$ to 0. We remark that this involves simple FNN constructions as done in other studies such as [36]. Informally, the way we construct $N_{\text{msg}}$ is to filter the correct temporal information. Otherwise, $\Diamond\psi$ is handled as seen in Theorem 6.

Correctness is again shown via induction over time and subformulae. The key insight here is that, while we do not directly evaluate formulas of the form $Q\psi$ with $Q \in \{\mathsf{Y}, \mathsf{P}\}$, the definition of $\mathcal{L}_2$ ensures that such formulas are of the form $Q\Diamond\psi$. Regarding $Q\Diamond\psi$, the interactions between $\phi$ and msg ensure that only the information from the previous (in the case of $Q = \mathsf{Y}$) or all previous (in the case of $Q = \mathsf{P}$) timepoints is considered in the aggregation. Other than this, the arguments are the same as in Theorem 6. $\square$

