# OpenReview forum: "The Logical Expressiveness of Temporal GNNs via Two-Dimensional Product Logics"
_NeurIPS.cc/2025/Conference — NeurIPS 2025 poster_

### Official Review · Reviewer_Hkih · 2025-06-30

**Clarity:** 3
**Significance:** 2
**Originality:** 3
**Rating:** 4
**Confidence:** 2

**Summary:**

This work proposes to use two-dimensional product logics to analyze the logical expressiveness of Temporal Graph Neural Networks (TGNNs).
The paper analyzes the expressiveness of three types of TGNNs: recursive, time-and-graph, and global TGNNs, using two types of logic: propositional temporal logic (PTL) and modal logic (K).
Theoretical results show that recursive TGNNs can express at least $PTL_{P,Y}\times K$, and that time-and-graph and global TGNNs can express fragments of $PTL_{P,Y}\times K$.

**Questions:**

- How could the theoretical results be utilized in practice? For example, could they be used to design more expressive TGNNs?
- Is there a reason for the use of truncated ReLU? I imagine those are not used in practice, and other activation functions such as sigmoid would be more common.
- Why do you only consider the operators $P$ and $Y$? Are there other past-time temporal logic operators that you could consider, such as ``always in the past''?
- More background details regarding some parts of the paper would be helpful, such as K^{\#} and time2vec.  While I understand they are given in the appendix, it would greatly help the readability of the paper if some details were included in the main text.

**Ethical Concerns:**

["NO or VERY MINOR ethics concerns only"]

**Final Justification:**

I appreciate the authors' response to the review.
I understand that this is a theory paper, but still have an impression that it is rather limited due to the lack of experimental progress.  I have already put a rather (weakly) positive score, which I feel is appropriate.

**Limitations:**

yes

**Quality:**

3

**Strengths And Weaknesses:**

Strengths:
- The paper is the first to analyze the logical expressiveness of TGNNs using two-dimensional product logics.
- The paper has strong theoretical results, showing which TGNNs can express what type of logic.
- Overall, the paper is well-written and easy to follow.

Weaknesses:
- The discussion on how the theoretical results could be utilized in practice is limited, despite the paper not including any experimental results.
- Reasoning behind the use of $PTL_{P,Y}$ is not stated clearly.

---

> ### Author Rebuttal · Authors · 2025-07-30
>
> Thank you for thoroughly reviewing our paper and posing several questions, which we happily address below. We focus on responding to the questions you asked.
>
> ---
>
> > How could the theoretical results be utilized in practice? For example, could they be used to design more expressive TGNNs?
>
> Exactly! We highlight three key uses of our research in practice:
>
> 1. It guides model selection, as some TGNN architectures are by design better suited for certain tasks than others.
> 2. Understanding the limitations of existing TGNN architectures, as initiated with our study, guides the design of more expressive types. However, practitioners must determine which capabilities are desirable.
> 3. Connecting TGNNs with logic paves the way for formal verification and interpretation of these models, for example, by extracting logical formulas from them.
>
> However, you're right that we could emphasise these key contributions even more. We plan to do so in the overview section of our results in a potential final version.
>
> > Is there a reason for the use of truncated ReLU? I imagine those are not used in practice, and other activation functions such as sigmoid would be more common.
>
> Please notice that all our established results also hold for the standard ReLU, as these can replicate the behaviour of truncated ReLU with only a linear increase in complexity.
>
> We have focused on truncated ReLU primarily to simplify proofs. Just as a sidenote: that it is a standard approach in papers relating the expressive power of GNNs with logics as those we considered as related work.
>
> > Why do you only consider the operators P and Y? Are there other past-time temporal logic operators that you could consider, such as ``always in the past''?
>
> Thank you for this comment! Actually our results apply also to "always in the past" operator.
>
> This is because "always in the past'' is expressible using negation and operator P. In particular, "always in the past $\varphi$'' is equivalent to: $\neg P \neg \varphi$. For simplicity of presentation and to make proofs easier (less operators to consider), we used the common approach to not include temporal operators which can expressed using combination of others.
>
> Please observe that for the same reason our presentation of the modal logic K mentions only $\Diamond$ among modal operators. However, the logic also allows to express the dual opeator $\Box\varphi$, as it is expressible as $\neg \Diamond \neg \varphi$.
>
> We will emphasise the above in the paper, which we believe can strengthen our results. Thank you for that!
>
>
> > More background details regarding some parts of the paper would be helpful, such as K^{#} and time2vec. While I understand they are given in the appendix, it would greatly help the readability of the paper if some details were included in the main text.
>
> Thank you for the comment. Limited amount of space requires moving some parts to the appendix, and it is the only reason why we have not provided full description of the above mentioned. We aim for a better explanation / intuitions about them in the revised version of the paper, especially Section 5.

---

> > ### Comment · Reviewer_Hkih · 2025-08-04
> > **Reply to rebuttal**
> >
> > Thank you for the clarification regarding the potential future works, as well as the clarification regarding logics used.
> > I will keep my score.

---

### Official Review · Reviewer_PQun · 2025-07-02

**Clarity:** 3
**Significance:** 3
**Originality:** 3
**Rating:** 4
**Confidence:** 3

**Summary:**

The paper presents an expressivity investigations for a temporal extension of Graph Neural Networks (GNN) hereinafter referred to as TGNNs. Instead of simply dealing with the topology of a graph as in GNNs, the model adds a temporal dimension considering a sequence of graphs marked by a timestamp. The paper recall the definition of GNN and then presents three different ways of extending their semantics to the temporal dimension given rise to the three classes of Recursive TGNNs, Time-and-graph TGNNs and Global TGNNs. The different way for including the temporal information (basically the way the embeddings on previous graphs are used to fix the embedding in the following ones) affects the expressiveness of corresponding classes. Following an idea  already successfully adopted in the literature for GNNs the expressiveness is compared by using a logical approach: which properties (expressed by a suitable logic) can be recognized by the considered class. In particular the logic exploited in the paper is a suitable product combination of a standard Past Linear Time Logic (for the temporal dimension) and the K logic (but also a fragment of CTL would suit as well) for the spatial dimension. The main result is that the logic is subsumed by Recursive TGNNs but not by Time-and-graph TGNNs and Global TGNNs. In addition, for by Time-and-graph TGNNs and Global TGNNs, the paper propose fragments of the product logic which are respectively subsumed.

**Questions:**

Page 3. Definition of Temporal Graph. The sequence of graphs in a temporal graph shares the same set of nodes. No constraint is required on the set of edges admitting different structures over the same set of nodes. Authors should spend some word on this point.
The logic K used in combination with PLTL allows only to express a modality for neighborhood. What would be the effects if a modality for reachability were considered (something like EF in CTL)?
The weakness of Global TGNNs seems to be the fact that they are not able to focus on specific instants or intervals on the past. Is there any significant class able to interact in more selective way with the past (not only the immediately preceding instant)?

**Ethical Concerns:**

["NO or VERY MINOR ethics concerns only"]

**Final Justification:**

I thank  the authors for their detailed discussions in the rebuttal. Indeed, there are some interesting future work directions opening up. As these discussions do not affect my evaluation of the paper, I will keep my score.

**Limitations:**

yes

**Quality:**

3

**Strengths And Weaknesses:**

Strength.  From the theoretical viewpoint, the paper focus on a relevant and interesting topic extending the expressiveness investigation on the temporal aspects of Timed GNN. The investigation provides an initial picture on the topic. Some aspects in the adopted technique are original (e.g. the adoption of the product logic). The technical machinery is appropriate and seems to be sound. The paper is clearly written even if some technical definitions needed to understand the paper are confined to the appendix.
Weakeness. Some results seems to immediately derive from corresponding results of the untimed case (e.g. Theorem 2 concerning presburger quantification). Quoting from the abstract: “These results  yields the first logical characterizations of temporal GNNs”.  Usually to have a “characterization” one has to prove that two formalisms are equivalent in some sense, so that claim should be removed. The correct claim is that stated by the title. If there had been a technical characterization the results would have assumed a much greater theoretical significance. The authors should explain more convincingly the usefulness of the temporal dimension.

---

> ### Author Rebuttal · Authors · 2025-07-30
>
> Thank you for taking the time to thoroughly review our paper and posing the insightful questions that we address below. We focus on responding to your specific queries below.
>
> ---
>
> > Definition of Temporal Graph. The sequence of graphs in a temporal graph shares the same set of nodes. No constraint is required on the set of edges admitting different structures over the same set of nodes. Authors should spend some word on this point.
>
> We will add a clarification of these assumptions in the revised version of the paper. We appreciate your attention to this matter!
>
>
> > The logic K used in combination with PLTL allows only to express a modality for neighborhood. What would be the effects if a modality for reachability were considered (something like EF in CTL)?
>
> Interesting point! Our study focuses on TGNNs, where the static graphs or snapshots of a temporal graph are processed by message-passing neural networks (MPNNs). In their standard, local-only form, these can only access the local neighbourhood of a node (the size of which depends on the depth of the MPNN). Thus, the exact classes of TGNNs we considered cannot capture modalities that express reachability.
>
> To capture "reachability", these MPNN components must somehow be able to access global information. This could be achieved by using recursive MPNN components, which have no fixed depth, or to some degree by MPNN components that allow for global access, sometimes referred to as global aggregation or global readout. Both kinds of MPNN exist (see [1, 2]), but are to the best of our knowledge not yet considered in the context of TGNNs. Similarly, a large part of TGNNs do not use MPNN-esque components to process static or temporal aspects of temporal graphs, but use attention mechanisms (see, for example, [3]). These may also capture modalities requiring global information.
>
>
> >The weakness of Global TGNNs seems to be the fact that they are not able to focus on specific instants or intervals on the past. Is there any significant class able to interact in more selective way with the past (not only the immediately preceding instant)?
>
> Your questions address the core challenge of navigating the TGNN landscape: it can be quite difficult to discern which models exist and how they differ in their capabilities. (To tackle this problem, we propose this study as a starting point to understand these differences with formal rigour.)
>
> Regarding global TGNNs, we note that certain types may indeed be capable of doing exactly what you are asking for. This necessitates that the way temporal information is combined with label information (in our paper realised by the $\circ$ operation and time function $\phi$) allows for Boolean-esque capabilities. Specifically, all label information corresponding to timepoints outside the interval we are interested in should be "ignored". In typical MPNN components, this is equivalent to mapping them to the all-zero vector, since aggregation is usually performed via summation. However, this assumes very specific conditions regarding the instantiation of the class of global TGNNs, namely $\circ$ and $\phi$, and it remains to be seen whether these conditions occur naturally in training processes. Nevertheless, these kinds of architectures generally permit the capabilities you are asking about.
>
>
>
> ---
> - [1] Veeti Ahvonen, Damian Heiman, Antti Kuusisto, Carsten Lutz:
> Logical characterizations of recurrent graph neural networks with reals and floats. NeurIPS 2024
> - [2] Michael Benedikt, Chia-Hsuan Lu, Boris Motik, Tony Tan:Decidability of Graph Neural Networks via Logical Characterizations. ICALP 2024
> - [3] Emanuele Rossi, Ben Chamberlain, Fabrizio Frasca, Davide Eynard, Federico Monti, Michael M. Bronstein:
> Temporal Graph Networks for Deep Learning on Dynamic Graphs. CoRR abs/2006.10637 (2020)

---

> > ### Author Response · Authors · 2025-08-06
> >
> > Thank you once again for your thoughtful and detailed feedback. Your questions touched on some of the most fundamental and exciting challenges in the TGNN landscape, and we truly appreciated the chance to engage with them.
> >
> > If anything remains unclear or if you would like to discuss certain aspects further, we would be more than happy to respond. We are grateful for any opportunity to help clarify and highlight the value of our contribution.

---

### Official Review · Reviewer_kogE · 2025-07-03

**Clarity:** 3
**Significance:** 2
**Originality:** 3
**Rating:** 4
**Confidence:** 4

**Summary:**

In this paper, the authors study the expressiveness of temporal graph
neural networks (TGNNs).

A GNN is defined by a sequence of aggregation and combination
functions that are applied to a graph G = (V, E, c), where c is a
function that maps each vertex in V to a vector in R^k. A temporal
graph is defined as (G_1, t_1), (G_2, t_2), ..., (G_n, t_n), where
each t_i is a timestamp and t_1 < t_2 < ... < t_n. Then, a TGNN is an
extension of a GNN that takes as input a temporal graph (G_1, t_1),
(G_2, t_2), ..., (G_n, t_n), processes each graph G_i as a GNN, but
assigns to each node u in (G_{i+1}, t_{i+1}) a label that is defined
as the concatenation of the label of u in G_{i+1} with the resulting
label from the execution on G_i.

The paper considers different classes of TGNNs, with the class of
recursive TGNNs being the most expressive. To study the expressiveness
of these classes, the paper considers different product logics, the
most expressive being defined as the product of path temporal logic
PTL_{P,Y}, with operators "yesterday" (Y) and "sometime in the past"
(P), with the modal logic K. Then, the authors prove that each formula
in PTL_{P,Y} × K can be expressed by a recursive TGNN. Moreover, the
authors consider less expressive TGNNs and identify fragments of
PTL_{P,Y} × K that are and are not contained in them. As a corollary
of these results, they provide separations in terms of expressive
power between the different classes of TGNNs considered in the paper.

**Questions:**

Q1 What are the complexities of the translations presented in the
paper? This information is important, for example, to determine
whether a TGNN is exponentially more succinct than a product logic.

**Ethical Concerns:**

["NO or VERY MINOR ethics concerns only"]

**Final Justification:**

The answer of the authors is only partially satisfactory, in particular as they do not comment on the fact that the paper does not identify any logic that exactly captures the expressive power of any of the classes of TGNNs studied. I keep my score.

**Limitations:**

Yes

**Paper Formatting Concerns:**

No formatting issues

**Quality:**

3

**Strengths And Weaknesses:**

Strengths:

S1 Understanding the expressive power of a TGNN is a fundamental
problem. In particular, understanding what cannot be expressed by a
class of TGNNs helps to understand not only the limitations of such an
architecture but also to identify what needs to be added to the
architecture to increase its expressive power.

S2 The paper provides some non-trivial results about the
expressiveness of some classes of TGNNs, which help in understanding
the expressiveness of these classes.

Weaknesses:

W1 The inexpressibility results presented in the paper are
interesting, as they help us understand what a class of TGNNs cannot
express. However, the authors do not make an effort to explain such
inexpressibility results in a way that the reader can clearly
understand the limitations of a class of TGNNs.

W2 The paper does not provide convincing evidence that product logics
are the right formalism to study the expressive power of TGNNs. In
particular, they do not clearly explain why the expressiveness of
TGNNs cannot be studied in terms of temporal extensions of first-order
logic.

W3 The paper does not identify any logic that exactly captures the
expressive power of any of the classes of TGNNs studied. Notice that
this has been done for (non-temporal) GNNs.

---

> ### Author Rebuttal · Authors · 2025-07-30
>
> Thank you for dedicating your time to providing a review of our paper. We focus addressing your direct question.
>
> ----
>
> > What are the complexities of the translations presented in the paper? This information is important, for example, to determine whether a TGNN is exponentially more succinct than a product logic.
>
> You are absolutely right that it is an interesting information to include in the paper. Please notice that all our translations are constructive and, thus, open to analyse their complexity.
>
> In particular, it is given that all of them are feasible in polynomial time. Indeed, all of our translations are based on enumeration of all (linearly many) subformulas of the formula to be translated, and then creating a single layer of a TGNN (or of its MPNN component, to be more specific) per subformula. This leads to a polynomial time procedure.
>
> Thank you for this comment! We will ensure to clarify and emphasise it in the revised version of the paper. We are certain that this observation will strengthen our results indeed.

---

> > ### Author Response · Authors · 2025-08-06
> >
> > Thank you again for your time and for raising this helpful point regarding the complexity of our translations. We hope our clarification was useful and helped strengthen the contribution.
> >
> > If there are any further questions or remarks, we would be happy to follow up. We’d greatly appreciate the opportunity to further discuss the work and ensure that its potential is fully understood

---

### Official Review · Reviewer_v7Dk · 2025-07-03

**Clarity:** 3
**Significance:** 2
**Originality:** 3
**Rating:** 4
**Confidence:** 3

**Summary:**

This paper investigates the expressive power of Temporal Graph Neural Networks (TGNNs) using two-dimensional product logics, which combine temporal and modal logics. The authors characterize the logical expressiveness of three classes of TGNNs—recursive, time-and-graph, and global TGNNs—by connecting them to fragments of these logics. They show that recursive TGNNs can express all properties definable in the product logic, while time-and-graph and global TGNNs are limited to specific fragments due to constraints on how temporal and spatial operators interact. These results provide foundational insights into the capabilities of different TGNN architectures and establish relative expressiveness relationships between them.

**Questions:**

1. How can the proposed method be extended to provide edge-specific or global confidence scores to enhance interpretability?
2. What is the time complexity (or computational cost) of extracting logical rules from a TGNN?

**Ethical Concerns:**

["NO or VERY MINOR ethics concerns only"]

**Final Justification:**

Thank you to the authors for providing discussions on several issues in the rebuttal. I believe these directions are worth further exploration in future work. However, as these discussions do not affect my evaluation of the paper, I have decided to maintain my rating.

**Limitations:**

The discussion on limitations, while present, could be more detailed, particularly regarding the assumptions and their impact on real-world applications.

**Quality:**

3

**Strengths And Weaknesses:**

## Strengths

1. The paper presents a novel and rigorous logical characterization of temporal graph neural networks (TGNNs) using two-dimensional product logics, effectively bridging a gap in the literature.
2. The theoretical results are well-supported by detailed proofs and clearly defined concepts, enhancing the paper’s credibility and rigor.
3. The work provides practical insights into the expressive power of different TGNN architectures, which could inform model selection for specific tasks.

## Weaknesses

1. The proposed method can derive logical interpretations for TGNNs but does not provide edge-specific or global weights (e.g., confidence scores), limiting interpretability at a granular level.
2. The paper lacks empirical validation, which could have strengthened the practical applicability of its theoretical findings.mpirical validation, which could have strengthened the practical relevance of the theoretical findings.
3. The technical complexity of the proofs and logical frameworks might limit accessibility for readers without a strong background in formal logics.

---

> ### Author Rebuttal · Authors · 2025-07-30
>
> Thank you for taking the time to review our paper. We will focus on addressing your specific questions.
>
> ---
>
> > How can the proposed method be extended to provide edge-specific or global confidence scores to enhance interpretability?
>
> You are correct that our proposed method aims not only to characterise the expressive capabilities of TGNNs, but also to pave the way for rigorous verification or interpretation methods based on logic. Combining interpretability methods, such as computing a global confidence score, with our approach of deriving logical characterisations seems to require connecting TGNNs with tailored "less-classical" logics, presumably those that include metric as well as probabilistic expressions.
>
> We can see several interesting research directions that could address this: (i) focus on establishing further characterisations (other classical temporal logics, other TGNN models, etc.), (ii) investigate verification/interpretation methods based on these, such as extracting equivalent formulas from TGNNs as surrogates; and (iii, your idea!) then proceed to combine logics with existing interpretation methods.
>
> > What is the time complexity (or computational cost) of extracting logical rules from a TGNN?
>
> Please notice that our proofs are constructive, as they provide algorithms for translating one formalism into another. However, there is no guarantee that our algorithms are optimal. Establishing tight computational complexity of translations between GNNs and logics is a very interesting task indeed. However, it is challenging, and so far it has not been fully addressed even in the conceptualy simpler case of static GNNs.
>
> However, some results for static GNNs are known, and there is an ongoing research in this direction. Assume we are interested in finding a formula $\varphi$ which exactly captures the classification of a static GNN $M$. Naive translations usually lead to an exponential size of $\varphi$. See for example [1]. However, there are substantial efforts in finding subclasses of static GNNs or certain verification problems, for which the translation is polynomial (like in [2]) or where verification is manageable despite a high worst-case complexity, see [3]. Solving the problem for static GNNs seems to be essential to make a progress for TGNNS, but it is reasonable to assume that similar research will be soon addressed also for TGNNs.
>
> Furthermore, if we assume that the formula $\varphi$ does not perfectly align with $M$ and only corresponds to it to a certain, but sufficient degree, then the time complexity might indeed be polynomial. This, however, is more related to approximation of one formalism with another, which we find as another very interesting direction!
>
>
> ---
> - [1] Michael Benedikt, Chia-Hsuan Lu, Boris Motik, Tony Tan:Decidability of Graph Neural Networks via Logical Characterizations. ICALP 2024
> - [2] Pierre Nunn, Marco Sälzer, François Schwarzentruber, Nicolas Troquard:
> A Logic for Reasoning about Aggregate-Combine Graph Neural Networks. IJCAI 2024
> - [3] Marco Sälzer, François Schwarzentruber, Nicolas Troquard:
> Verifying Quantized Graph Neural Networks is PSPACE-complete. CoRR abs/2502.16244 (2025)

---

> > ### Author Response · Authors · 2025-08-06
> >
> > Thank you again for taking the time to read our work and for posing such thoughtful questions. We truly appreciate the opportunity to reflect on them during the rebuttal.
> >
> > If there are any remaining concerns or open points, we would of course be very happy to clarify or elaborate further. We would be grateful for any opportunity to help fully convey why we believe our contribution offers a meaningful step toward a deeper understanding of TGNN expressiveness and its implications for verification and interpretability.

---

### Note · Authors · 2025-08-12

We thank all reviewers and the AC for their time and valuable feedback.

All reviews of our submission were positive, and we believe our earlier responses have addressed the points and questions raised, such that no further follow-up discussion was necessary.

As a summary of the discussion period, please let us mention how we have addressed each reviewer’s key points:
**v7Dk**: We discussed how our framework could be extended with non-classical logics to support global/edge confidence scoring, and outlined complexity considerations with reference to related static GNN results.
**kogE**: We explained that all our translations are constructive and run in polynomial time.
**PQun**: We clarified the definition of temporal graphs; discussed reachability as a potential modality, selective past-interaction, and conditions under which these could be supported.
**Hkih**:  We highlighted practical uses of our results (model selection, guiding architecture design, enabling verification); explained choice of truncated ReLU; clarified temporal operator selection; and committed to adding more intuition on background notions.

We believe these clarifications directly address the reviewers’ questions and strengthen the connection between our theoretical contributions on TGNN expressiveness and their practical implications.

---

### Decision · Program_Chairs · 2025-09-17

**Decision:**

Accept (poster)

**Comment:**

(a) Summary of scientific claims and findings

This paper studies the expressive power of temporal graph neural networks (TGNNs) by connecting them to two-dimensional product logics, combining propositional temporal logic (PTL) and modal logic K. The main result is that recursive TGNNs can capture all properties definable in PTL × K, while other classes such as time-and-graph TGNNs and global TGNNs are limited to syntactically constrained fragments. This provides the first logical characterizations of TGNNs and clarifies their relative expressiveness.

(b) Strengths

- Provides a rigorous and novel theoretical framework linking TGNNs to product logics.
- Establishes clear separations in expressive power between different TGNN architectures.
- Proofs are constructive, with translations running in polynomial time.
- Potentially useful for guiding model selection, architecture design, and verification.

(c) Weaknesses

- No empirical validation or experiments, limiting the practical impact for applied audiences.
- Some results (e.g., fragments derivable from static GNN expressiveness) could be seen as incremental extensions.

(d) Reasons for decision

I recommend acceptance as a poster. The theoretical contribution is sound, original, and relevant to the NeurIPS theory community. The paper advances our understanding of TGNN expressiveness, and while not groundbreaking enough for a spotlight or oral (due to lack of experiments, incremental nature of some parts, and partial scope), it is nonetheless a meaningful and rigorous step in an emerging area.

(e) Discussion and rebuttal period

During the discussion, reviewers raised concerns about interpretability extensions, computational complexity, the choice of formalism, missing exact characterizations, assumptions on temporal graphs, and practical relevance of the results. The authors clarified that their translations are constructive and polynomial time, explained architectural limitations tied to reachability, justified operator and activation choices, and outlined how their work connects to verification and model design. While the rebuttal effectively addressed technical questions, broader conceptual and scope-related concerns remained. Overall, reviewers converged on borderline positive evaluations, and the paper is judged solid enough for poster acceptance.